

# Measurement report: Formation and brownness of aqueous secondary organic aerosol from the aged biomass-burning emissions in the Sichuan Basin, China

Chao Peng[1,2,3,4], Yan Ding[4], Zhenliang Li[1,2,3], Tianyu Zhai[4], Xinping
Yang[4], Mi Tian[6], Yang Chen[5], Xin Long[5], Haohui Tang[6], Guangming
Shi[7], Liuyi Zhang[8], Kangyin Zhang[8], Fumo Yang[7], and Chongzhi Zhai[1,2,3]
[1]Chongqing Academy of Ecology and Environmental Sciences, Chongqing, 401336,
China
[2]Chongqing Branch Academy of Chinese Research Academy of Environmental
Sciences, Chongqing, 401336, China
[3]Chongqing Key Laboratory of Urban Atmospheric Environment Observation and
Pollution Prevention, Chongqing, 401336, China
[4]Chinese Research Academy of Environmental Sciences, Beijing 100012, China
[5]Chongqing Institute of Green and Intelligent Technology, Chinese Academy of
Sciences, Chongqing, 400714, China
[6]College of Environment and Ecology, Chongqing University, Chongqing, 400045,
China
[7]College of Carbon Neutrality Future Technology, Sichuan University, Chengdu,
610065, China
[8]Chongqing Three Gorges University, Wanzhou, 404000, China
**Correspondence:** Chao Peng (pengchao0623@sina.com) and Chongzhi Zhai
(czz66818@sina.com)





**Abstract.** Secondary organic aerosol (SOA) formed via complex chemical
mechanisms was the major contributor to atmospheric aerosol pollution and climate
forcing worldwide. The aqueous-phase oxidation was an important pathway for SOA
formation and the aqueous SOA (aqSOA) exhibited absorption properties across
ultraviolet to visible range. Here, we reported the direct ambient observation of SOA
formation and absorption properties in the aqueous phase from the Sichuan Basin,
China. Considerable aqSOA was originated from the aged biomass-burning emissions
via aqueous-phase reactions instead of photo-chemical reactions under high aerosol
liquid water content (ALWC) conditions, especially during the polluted period. The
substantial impact on brown carbon (BrC) absorption from SOA was observed from
370 nm to 660 nm (27.5%−43.2%). This study highlighted the significant contribution
of aqSOA formation from aged biomass-burning emissions to the BrC budget and
absorption, especially at night. The mean aerosol absorption Ångström exponents
from 370 nm to 880 nm ($AAE_{370-880}$) was 1.95, higher than that observed in fresh and
photo-chemically aged biomass-burning emissions. This study revealed the aqSOA
formation and brownness from aged biomass-burning emissions and highlighted the
importance of aqueous-phase reactions on air quality and climate.
**Keywords:** Particulate matter; Secondary organic aerosol; Aqueous-phase oxidation;
Aged biomass-burning emissions; Brown carbon.



**1 Introduction**

Organic aerosol (OA) was the dominant component (20 to 90%) of atmospheric
aerosol with significantly implications for air quality and climate forcing (Jimenez et
al., 2009). Numerous field observations indicated that secondary OA (SOA), formed
by atmospheric oxidation of volatile organic compounds (VOCs) and primary OA
(POA), accounted for most of OA worldwide (Ervens et al., 2011; Huang et al., 2014;
Kourtchev et al., 2016). Recent results showed that aqueous-phase oxidation was an
important pathway for SOA formation and these SOA production (aqSOA) exhibited
absorption properties across ultraviolet (UV) to visible (Vis) range (Gilardoni et al.,
2016; Lim et al., 2010; McNeill 2015; Powelson et al., 2014; Sun et al., 2010).
However, the formation mechanisms and absorption properties of aqSOA were poorly
understood, hindering to improvement of air quality and reducing the uncertainties in
global climate estimations.
An increasing number of studies pointed toward aqSOA as a major SOA could
form in fogs, clouds, and aerosol water, and oxygenated VOCs (OVOCs) with large
water-soluble and low Henry's constant (i.e., methylglyoxal and glycolaldehyde) were
the important aqSOA precursors (Ervens et al., 2011; Ortiz-Montalvo et al., 2012; Tan
et al., 2012; Xu et al., 2022). A few laboratory studies investigated the levoglucosan
and phenolic species produced from biomass burning could also act as aqSOA
precursors (Yu et al., 2016; Zhao et al., 2014). Gilardoni et al. (2016) reported direct
ambient observations of aqSOA formation from biomass-burning emissions in fog
water and wet aerosol. Additionally, recent studies indicated that aqSOA with high



molecular weight (i.e., 4-ethylphenol) formed by aqueous-phase photochemical
oxidation showed strong light absorptivity within UV range (Herrmann et al., 2015;
Ye et al., 2019). Previous laboratory studies also demonstrated that aqSOA, such as
π-conjugated compounds and imidazole with C=N bonds produced by aldol
condensation and aqueous-phase carbonyl compound reactions respectively, would
strongly absorb light at near-UV (Drozd and McNeill, 2014; Kampf et al., 2012;
Nozière and Esteve, 2007; Powelson et al., 2014). Despite numerous studies reported
on the formation and optical properties of aqSOA, limited research on its ambient
observations hindered to better understand the role of aqSOA in atmospheric
chemistry and climate.
China experienced severe $PM_{2.5}$ pollution under the stagnant high-humidity
conditions, when SOA as the major component was originated from fossil fuel
combustion and biomass burning (Huang et al., 2014; Wang et al., 2016; Wang et al.,
2021; Xu et al., 2022). Field observations indicated that highly oxidized SOA could
form through aqueous-phase processing driven by acid-catalyzed oxidation (Meng et
al., 2020; Xu et al., 2017), and considerable aqSOA was formed from
biomass-burning OA (BBOA) and fossil-fuel OA via aqueous-phase reactions (Wang
et al., 2021; Zhao et al., 2019). A few laboratory studies found aqueous-phase
reactions were an important oxidation pathway for nitrophenol products (i.e.,
5-nitrovanillin and 4-nitroguaiacol) with strong UV absorption and higher formation
and transformation rates were observed in more acidic solutions (Kroflic et al., 2015;
Li et al., 2023; Pang et al., 2019; Yang et al., 2021). However, observations on aqSOA



formation and optical properties in China were limited and most research
concentrated on the North China Plain (NCP). Similar to NCP, the Sichuan Basin
(SCB) characterized by high humidity and frequent biomass burning was also the
main region with severe aerosol pollution in China (Tian et al., 2019; Wang et al.,
2018; Yang et al., 2011). Currently, few studies explored the dynamic evolution and
optical properties of aqSOA, and the knowledge of ambient aqSOA processing was
still limited in SCB. Therefore, a more detailed characterization of aqSOA formation
and optical properties was of great importance to reveal the key factors contributing to
haze formation.

Here a time-of-flight aerosol chemical speciation monitor (ToF-ACSM) and a

series of collocated instruments were used to characterize aqSOA dynamic evolution
from biomass burning under real ambient conditions in a typical city in SCB from
October 21 to November 23, 2022. We observed that the haze formation was largely
driven by BBOA and aqSOA. We demonstrated considerable aqSOA was originated
from the aged BBOA via aqueous-phase reactions. Finally, we further showed that
aqSOA produced from aged BBOA were strong UV absorption with positive radiative
forcing. These results revealed the aqSOA formation and brownness from aged
biomass-burning emissions and helped simulate the associated influences on
atmospheric chemistry and climate.



## 2 Methods

### 2.1 Sampling site

An intensive field campaign on the chemical and physical properties of aerosol was conducted at a site in a severe aerosol pollution city (Yongchuan, 29°21′25″ N, 105°54′6″ E) from October 21 to November 23, 2022. This was a typical urban site located in a parallel ridge-and-valley area between two megacities in SCB (Chongqing center and Chengdu). It was primarily influenced by multiple local emissions from traffic (arterial roads to the east 600 m and west 300 m) and a variety of residential sources (i.e., biomass burning and fossil fuel combustion). Measurements at the site were not interfered by neighboring buildings and helped understand the characteristics of haze pollution dynamic evolution.

### 2.2 Instrumentation

During the campaign, the non-refractory aerosol (NR-PM$_{2.5}$) species, including organics (Org), ammonium (NH$_4$), nitrates (NO$_3$), sulfates (SO$_4$), and chlorides (Chl), were measured on-line by ToF-ACSM (Aerodyne Research Inc.). Ambient aerosols were pumped into ToF-ACSM at a flow rate of 3 L min$^{-1}$ through a PM$_{2.5}$ cyclone (URG-2000-30ED) and a Nafion dryer (MD-110-48S, Perma Pure, Inc.) reducing the relative humidity to below 30%. The measurement principle was described in detail in the previous studies (Fröhlich et al., 2013; Ng et al., 2011). The ionization efficiency (IE) and relative ionization efficiency (RIEs) were regularly calibrated by a scanning mobility particle sizer with a differential mobility analyzer (SMPS 3081A, TSI) and a



condensation particle counter (CPC 3775, TSI). The comprehensive overview of the
operation and calibration procedures of ToF-ACSM could be found in Bao et al.

(2023).

A seven-wavelength Aethalometer (AE33, Magee Scientific) was used to

measure the aerosol light absorption ($Abs_\lambda$) and black carbon ($BC_\lambda$) mass
concentrations in real time at 370, 470, 520, 590, 660, 880, and 950 nm. The sampled
particles were dried by a Nafion dryer (MD-70024S-3, Perma Pure, Inc.) before
entering into AE33. The light attenuation coefficients were converted to $Abs_\lambda$ based
on the real-time compensation parameter, and the nonlinear loading effects of quartz
filters were dealt with on-line by the parallel measurements of attenuation values
(ATN1 and ATN2) (Coen et al., 2010; Drinovec et al., 2015). The scattering effects of
quartz filters were modified automatically by a fixed multiple scattering parameter
(2.14). Detailed measurement methods and principles of AE33 could be found in
Drinovec et al. (2015).

During the campaign, the gaseous species (including $NO_2$ and CO) were

continuously measured by gas analyzers (42i and 48i, Thermo Scientific), that were
maintained and calibrated weekly. Hourly meteorological parameters data including
temperature (T), relative humidity (RH) and $PM_{2.5}$ mass concentrations were obtained
on-line from the measurements at the National Environmental Monitoring Station,
which was close to our sampling site (http://www.cnemc.cn/).



**2.3 Data analysis**
**2.3.1 ToF-ACSM data analysis**
The raw mass spectra data measured by ToF-ACSM were analyzed using
Tofware v2.5.13 (Tofwerk AG) in Igor Pro 6.37 (WaveMetrics, Inc.). In accordance
with previous studies, the default RIEs values for Org, $NO_3$, and Chl were set to 1.4,
1.1, and 1.3, respectively (Canagaratna et al., 2007; Elser et al., 2016). The IE value
(236 ions $pg^{-1}$) and RIEs of $SO_4$ (1.2) and $NH_4$ (4.3) were estimated from the
calibrations of pure ammonium nitrate and ammonium sulfate, respectively. The
collection efficiency (CE) was set to 0.5 considering the minor influences of low RH
dried by Nafion dryer (< 30%), low fraction of ammonium nitrate in NR-$PM_{2.5}$ (31%),
and ammonium-rich conditions (Fig. S1) (Matthew et al., 2008; Middlebrook et al.,
2012; Zhao et al., 2019). Additionally, the strong correlation between NR-$PM_{2.5}$ and
$PM_{2.5}$ mass concentrations supported that the CE value was reasonable (Fig. S2).
The mass spectral matrix of OA for m/z 10−120 was analyzed by positive matrix
factorization (PMF) and multilinear engine (ME2) implemented with the SoFi 6.3
(Canonaco et al., 2013; Paatero 1999; Paatero and Tapper 1994). Briefly,
unconstrained PMF was used to determine the numbers and types of source factors,
then the restriction method ME2 was used to minimize PMF rotational ambiguity by
the *a*-values from 0 to 1 with a step of 0.1 (Wang et al., 2019; Zhong et al., 2021). The
ions data with signal-to-noise (S/N) lower than 0.2 were discarded, and those S/N
from 0.2−2 were downweighted by a factor of 2 (Bao et al., 2023). Finally, five OA
factors with function of the rotational parameter ($f_{peak} = 0$) were identified, including





BBOA, coal-combustion OA (CCOA), hydrocarbon-like OA (HOA), oxygenated OA
(OOA), and aqSOA. The details of OA source apportionment procedures were
described in SI Text S1.
**2.3.2 Aerosol liquid water content**

During the campaign, the aerosol liquid water content (ALWC) was estimated by

the ISORROPIA-II model based on the ammonium, nitrates, sulfates, and chlorides
mass concentrations from ToF-ACSM and the meteorological parameters (T and RH)
from National Environmental Monitoring Station (Fountoukis and Nenes, 2007). Here,
the forward type and metastable mode were used in the ISORROPIA-II model
(Hennigan et al., 2015). The thermodynamic equilibrium of the
$NH_4^+-SO_4^{2-}-NO_3^--Cl^--H_2O$ system was modeled and ALWC was then calculated.
**2.3.3 Light absorption measurements**

The $Abs_\lambda$ was divided into BC and brown carbon (BrC, a group of colored OA

compounds) absorption ($Abs_{\lambda,BC}$ and $Abs_{\lambda,BrC}$) ( $Abs_\lambda=Abs_{\lambda,BC}+Abs_{\lambda,BrC}$ ) and
characterized by the absorption Ångström exponents (AAE) (Laskin et al., 2015).
Here, $Abs_\lambda$ was determined dependent $BC_\lambda$ mass concentrations ($Abs_\lambda=BC_\lambda \times MAC_\lambda$).
We assumed the mass absorption cross-section of aerosols ($MAC_\lambda$) were 18.47, 14.54,
13.14, 11.58, 10.35, 7.77, and 7.19 $m^2$ $g^{-1}$ at 370, 470, 520, 590, 660, 880, and 950
nm, respectively (Drinovec et al., 2015; Zhu et al., 2017). Here, $Abs_{880}$ was sole from
BC and the following formula was used to determine $Abs_{\lambda,BC}$ values:
$Abs_{\lambda,BC}=Abs_{880}\times(\lambda/880)^{-AAE_{BC}}$ (Kirchstetter and Novakov, 2004; Moosmüller et al.,



2009). The AAE of BC ($AAE_{BC}$) value was obtained from the equality:
$AAE_{BC}=-\log(Abs_{880}/Abs_{950})\div\log(880/950)$ (Wang et al., 2021). Additionally,
$Abs_{\lambda,BrC}$ was caused by primary and secondary BrC light absorption ($Abs_{\lambda,BrC,pri}$ and
$Abs_{\lambda,BrC,sec}$). The $Abs_{\lambda,BrC,sec}$ value was calculated by a minimum R-squared (MRS)
method at each wavelength (Wang et al., 2019; Wu and Yu, 2016; Wu et al., 2024).
The detailed information of MRS method and $Abs_{\lambda,BrC,sec}$ estimation was provided in
SI Text S2.

The multiple linear regression (MLR) method was used to analyze the light

absorption of different OA component at each wavelength:
$Abs_{BrC}=a\times[OOA]+b\times[BBOA]+c\times[CCOA]+d\times[aqSOA]+e\times[HOA]$ (Qin et al., 2018;
Xie et al., 2019). The [OOA], [BBOA], [CCOA], [aqSOA], and [HOA] indicated the
mass concentrations of OA species; the a–e were constants, used to optimize the $Abs_{\lambda}$
of each OA component, and were equivalent to MAC values at each wavelength. Here,
the normalized mean bias (NMB), root mean square error (RMSE), and index of
agreement (IOA) were used to evaluate the performance of the MLR method (SI Text
S3) (Li et al., 2011). The IOA values of $Abs_{370,BrC}$ and $Abs_{470,BrC}$ (0.99 and 1.00)
exceeded 0.95. The slopes of the relationship between $Abs_{370,BrC}$ and $Abs_{470,BrC}$
measured by AE33 and estimated by MLR method were 0.81 and 0.96 (close to unity),
respectively. These results indicated a good agreement of $Abs_{370,BrC}$ between AE33
measurement and the MLR reconstruction.



**3 Results and discussion**
**3.1 General descriptions**
The temporal variations of $PM_{2.5}$ species concentrations, meteorological
parameters, $Abs_{370,BrC}$ and $MAC_{370,BrC}$ during the campaign were shown in Fig. 1. The
winds were weak with $0.3 \pm 0.2$ m s$^{-1}$ over the whole campaign, indicating the
atmosphere was in stagnant conditions. The total $PM_{2.5}$ (BC+NR-$PM_{2.5}$) mass
concentration ranged from 7.0 to 175.5 μg m$^{-3}$, with an average of $48.4 \pm 27.8$ μg m$^{-3}$
during the campaign. The average concentrations of Org, $NO_3$, $SO_4$, $NH_4$, Chl, and
BC were $24.1 \pm 18.1$, $8.3 \pm 6.2$, $6.2 \pm 3.4$, $5.2 \pm 2.7$, $0.2 \pm 0.1$, and $4.7 \pm 2.9$ μg m$^{-3}$,
taking up $46.6 \pm 10.7\%$, $17.7 \pm 8.0\%$, $13.2 \pm 4.4\%$, $11.2 \pm 2.7\%$, $0.3 \pm 0.2\%$, and 10.1
$\pm 5.5\%$ of total $PM_{2.5}$, respectively. Org constituted the largest fraction of total $PM_{2.5}$,
highlighting the importance of OA in $PM_{2.5}$ pollution in SCB (Bao et al., 2023; Wang
et al., 2018). Meanwhile, the high values of $Abs_{370,BrC}$ and $MAC_{370,BrC}$, ranging from
5.8 to 210.2 Mm$^{-1}$ ($42.4 \pm 28.5$ Mm$^{-1}$) and from 0.6 to 7.0 m$^2$ g$^{-1}$ ($2.1 \pm 0.9$ m$^2$ g$^{-1}$)
respectively, were observed during the campaign. It was worth noting that the $PM_{2.5}$
species and OA composition were substantially different in the polluted period (PP)
(BC+NR-$PM_{2.5}$ > 75 μg m$^{-3}$) and clean period (CP) (BC+NR-$PM_{2.5}$ ≤ 75 μg m$^{-3}$).
During the PP, the mass concentrations of BC+NR-$PM_{2.5}$ and OA were 102.3 $\pm$
26.9 and $57.4 \pm 22.5$ μg m$^{-3}$, 2.5 and 3.1 times that during CP, respectively. Compared
with other species, a significantly higher contribution of OA was observed during PP
(56.6%) than CP (46.6%) (Student's t-test, $p < 0.001$) (Fig. 2). Here, five OA factors
were identified by the PMF-ME2 model with detailed information in SI Text S1, and



the mass spectrum of these factors was shown in Fig. S3. Moreover, BBOA showed
significant correlations with $C_2H_4O_2^+$ and m/z 73 ($r^2 = 0.85, 0.80, p < 0.001$); CCOA
was strongly correlated with Chl and m/z 115 ($r^2 = 0.56, 0.48, p < 0.001$); HOA was
correlated with $NO_2$ and m/z 41 ($r^2 = 0.47, 0.59, p < 0.001$); OOA and aqSOA were
significantly correlated with $NO_3$, $NH_4$ ($r^2 = 0.77, 0.75, p < 0.001$) and $SO_4$, ALWC
($r^2 = 0.67, 0.85, p < 0.001$), respectively (Fig. S4). These results highlighted the result
of five OA factors was reasonable.

It should be noted that the contributions of BBOA and aqSOA to OA increased

from CP (31.7% and 12.6%) to PP (38.6% and 14.1%), respectively. Additionally,
significantly higher RH and ALWC were observed during PP (58.5 ± 12.4% and 69.4
± 30.3 µg m$^{-3}$) than CP (49.8 ± 8.9% and 37.1 ± 20.8 µg m$^{-3}$) ($p < 0.001$), but not
temperature ($p > 0.1$). The wind was 0.32 ± 0.18 m s$^{-1}$ during CP, 1.3 times that
during PP. These results indicated that the atmosphere was in a stagnant state with
relatively high RH and ALWC during PP, which might lead to the largely different
sources and chemical processing of OA during CP and PP. As shown in Fig. 2, the
obvious diurnal variation of OA concentration with one peak (82.7 µg m$^{-3}$) appeared
at 12:00 local time (LT) in the daytime was exhibited during PP, but 21:00 LT at night
during CP. Moreover, OA concentration rapidly increased at a rate of 7.8 µg m$^{-3}$ hr$^{-1}$
from 09:00 to 12:00 LT with a significant decrease of $NO_3$ during PP. Meanwhile,
BBOA and aqSOA concentrations showed similar diurnal patterns to OA
concentration with high values in the daytime and rapidly increased from 09:00 to
12:00 LT during PP. Previous research indicated that aqSOA spectrum showed higher



m/z 29 (CHO$^+$) than other OA factors (Gilardoni et al., 2016; Meng et al., 2020; Wang
et al., 2021). During PP, the peaks of C$_2$H$_4$O$_2^+$ and m/z 29 concentrations, tracer ion
fragments of BBOA and aqSOA, were observed at 12:00 LT (1.2 μg m$^{-3}$) and 13:00
LT (4.3 μg m$^{-3}$), respectively. Additionally, the correlation between ALWC and
aqSOA concentration ($r^2$ = 0.86, $p$ < 0.001) was stronger than BBOA concentration ($r^2$
= 0.58, $p$ < 0.001), and both ALWC and aqSOA concentration peaks were observed at
13:00 LT, earlier than BBOA concentration peak (12:00 LT), supporting that ALWC
might play a significant role in the chemical processing of BBOA and aqSOA during
PP.

In summary, these results suggested that OA was the dominant component of

PM$_{2.5}$, especially during PP in SCB. During PP, BBOA and aqSOA played important
roles in increasing OA concentration in the daytime. Additionally, considerable
aqSOA could be formed from BBOA via aqueous-phase reactions under high ALWC
in the daytime during PP, which might be related to the frequent fog events and
BBOA emissions in the harvest season − autumn − in SCB (Bao et al., 2023; Chen et
al., 2017; Chen et al., 2019; Tao et al., 2014). To further explore aqSOA formed from
BBOA via the aqueous-phase reactions, the next section would discuss the field
observation of aqSOA from biomass-burning emissions.





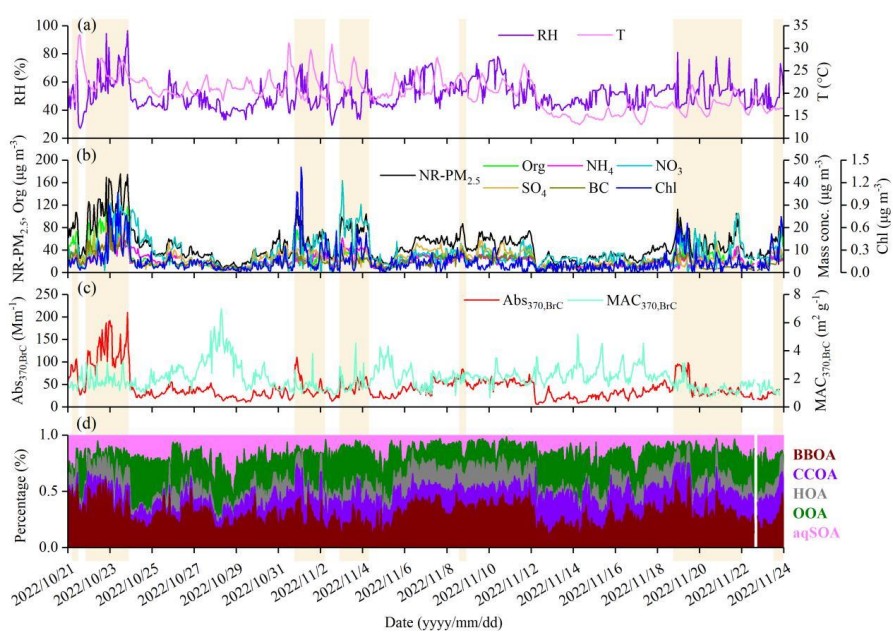


**Figure 1.** Time series of **(a)** RH and T, **(b)** NR-PM$_{2.5}$ species measured by ToF-ACSM and BC, **(c)**

Abs$_{370,BrC}$ and MAC$_{370,BrC}$, and **(d)** mass fraction of OA factors during the campaign. The pollution

period (BC+NR-PM$_{2.5}$ > 75 µg m$^{-3}$) is highlighted by the shaded areas.

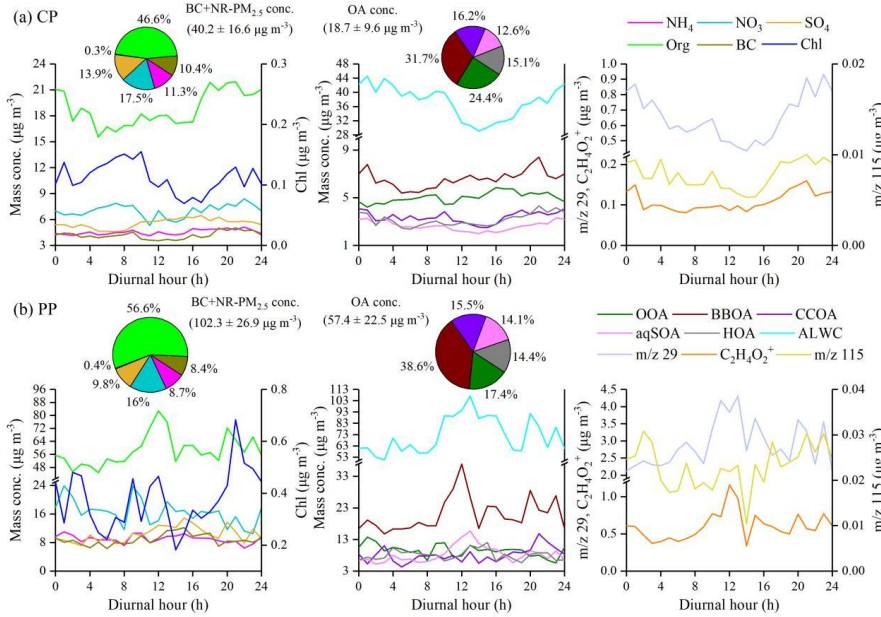




**Figure 2.** Diurnal variations of PM$_{2.5}$ species, BC, OA factors, m/z 29, m/z 115, and C$_2$H$_4$O$_2^+$
mass concentrations during **(a)** clean period (CP) (BC+NR-PM$_{2.5}$ < 75 μg m$^{-3}$) and **(b)** polluted
period (PP) (BC+NR-PM$_{2.5}$ > 75 μg m$^{-3}$). The pie charts in the left side of **(a)** and **(b)** show the
average mass contributions of different chemical compositions to BC+NR-PM$_{2.5}$ during CP and PP,
respectively. Meanwhile, the average mass contributions of OOA, BBOA, CCOA, aqSOA, and
HOA in OA are shown in the pie charts in the middle of **(a)** and **(b)**, respectively.
**3.2 Biomass-burning emissions as precursors for aqSOA**

280   Fig. 3 showed the relationship between the mass fraction (%) of aqSOA in total

PM$_{2.5}$ and ALWC during the campaign. There was a strong correlation with increasing
the contribution of aqSOA at the high $f_{29}$ values (normalized mass spectrum signal at
m/z 29) ($r^2$ = 0.64, $p$ < 0.001). It was important to note that the aqSOA factor showed
significantly higher $f_{29}$ and $f_{60}$ values (normalized mass spectrum signal at m/z 60)
(0.167 and 0.015) and lower $f_{44}$ value (normalized mass spectrum signal at m/z 44)
(0.097) than the OOA factor (0.017, 0.002, and 0.181), respectively (Fig. S3).
Moreover, both aqSOA concentrations and $f_{29}$ were well correlated with ALWC ($r^2$ =
0.85, 0.73, $p$ < 0.001), and the BBOA factor was located in a similar region with
aqSOA factor. These results were similar to aqSOA observed in Italy and Beijing
(Gilardoni et al., 2016; Zhao et al., 2019), indicating that considerable aqSOA could
be formed from biomass-burning OA via aqueous-phase reactions in SCB.
Additionally, a strong anticorrelation between the mass fraction of fossil-fuel related
OA components (sum of CCOA, HOA and OOA) and ALWC at the high $f_{29}$ values



was also observed ($r^2$ = 0.48, $p$ < 0.001), indicating that aqSOA might also be
production by aqueous-phase reactions of fossil-fuel related OA components,
consistent with previous research (Wang et al., 2021).
Fig. 4 showed the relationships between ALWC and OA factors or $f_{29}$ during the
campaign. Five OA factors mass concentrations increased with the increase of ALWC.
However, compared with other OA factors, aqSOA and BBOA significantly increased
from 1.1 and 4.9 µg m$^{-3}$ to 5.2 and 10.8 µg m$^{-3}$ when 20 µg m$^{-3}$ < ALWC < 60 µg
m$^{-3}$, respectively. It should be noted that only aqSOA concentrations were even
enhancement under high ALWC conditions (> 100 µg m$^{-3}$), which might be related to
more water-soluble organic species (i.e., glyoxal and methylglyoxal) were formed and
further formed aqSOA via aqueous-phase reactions in aerosol liquid water (Carlton et
al., 2007; Ervens et al., 2011; Tan et al., 2012). Based on the direct observation of
aqSOA, Gilardoni et al. (2016) also found that aqSOA such as guaiacol dimer
($C_{14}H_{14}O_4^+$) could be formed from aged biomass-burning emissions at both in fog
water and in wet aerosol, especially under high ALWC conditions. As shown in Fig.
4b, the mass fraction of aqSOA showed significant enhancement from less than 5% at
ALWC < 20 µg m$^{-3}$ to 17−22% at ALWC > 60 µg m$^{-3}$ with a corresponding decrease
in OOA, although POA and SOA contributions were fairly constant across different
ALWC levels (58−68% and 32−42%). This result suggested a more intensive
formation of aqSOA than OOA via aqueous-phase reactions, although aqSOA might
be also formed from OOA, consistent with the recent research in northwest China
(Zhao et al., 2019; Zhong et al., 2021). Additionally, the increasing $f_{29}$ (CHO$^+$) from



0.010 to 0.227 as a function of ALWC was observed during the campaign (Fig. 4b).
The values of $f_{29}$ significantly increased from 0.055 to 0.210 when ALWC increased
from 60 μg m$^{-3}$ to 100 μg m$^{-3}$ ($p < 0.001$), consistent with OA mass concentrations
(13.2−109.1 μg m$^{-3}$) during the campaign (Fig. 4b). According to the laboratory
analysis of organic standards, previous research found that the spectra of standard
organic species without alcohol group showed low $f_{29}$ (< 0.05), while high $f_{29}$ values
(0.05–0.15) were found for polyols and species with non-acid OH groups produced
from biomass-burning emissions (Canagaratna et al., 2015; Gilardoni et al., 2016;
Zhao et al., 2014). This further highlighted the potential formation of organic
compounds with hydroxyl groups (i.e., glyoxal and methylglyoxal) under high ALWC
conditions. Overall, these results pointed to the fact that the observed aqSOA could be
formed from biomass-burning emissions via aqueous-phase reactions, reinforcing the
BBOA role in increasing PM$_{2.5}$ mass concentration.

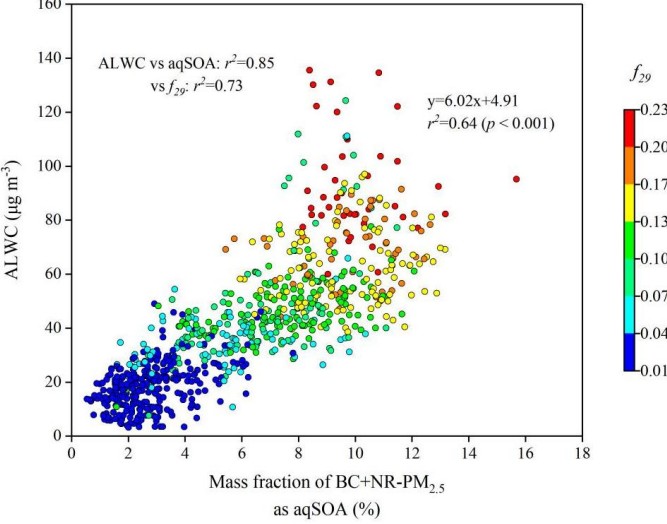


**Figure 3.** Scatter plot of the mass fraction of aqSOA in BC+NR-PM$_{2.5}$ versus ALWC colored by





the $f_{29}$ (normalized mass spectrum signal at m/z 29) during the campaign. $f_{29}$ (mainly CHO$^+$) is a
tracer for alcohol compounds and used to monitor the aqueous-phase oxidation of organic
compounds (i.e., glyoxal).

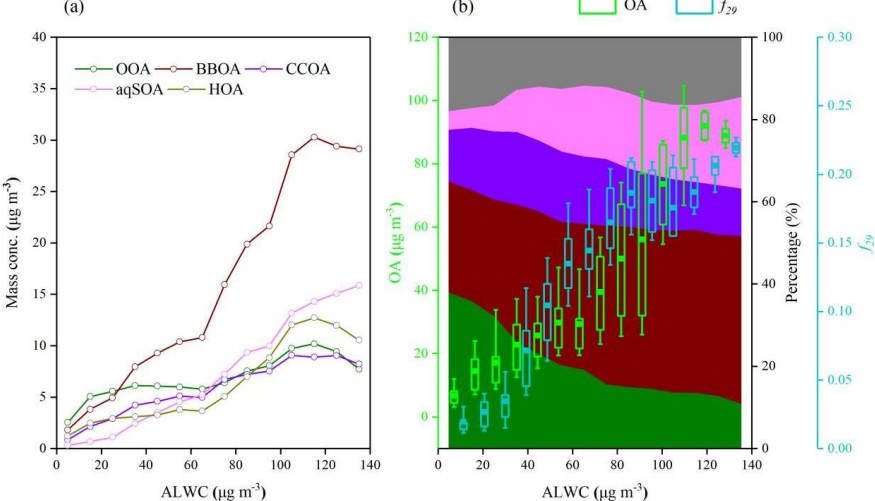

**Figure 4.** Variations of **(a)** OA factors mass concentrations, and **(b)** OA mass concentrations, $f_{29}$ (a

tracer for alcohol compounds), and mass fraction of OA factors as a function of ALWC. The data
were grouped into different bins according to a 10 μg m$^{-3}$ increment of ALWC.
To identify the formation of aqSOA and its precursors under different PM$_{2.5}$
pollution levels, the relationship between aqSOA and BBOA or OOA mass
concentrations with ion fragments tracers during CP and PP was performed,
respectively. As shown in Fig. 5a and c, aqSOA and BBOA concentrations increased
with the increase of ALWC in general during CP and PP. However, compared with
CP ($r^2 = 0.54$), the stronger positive correlation between aqSOA and BBOA
concentrations was observed during PP ($r^2 = 0.64$), and so were that between ALWC
and aqSOA or BBOA concentrations ($r^2 = 0.93$, 0.59, respectively). This supported



high BBOA concentrations were favorable for aqSOA formation, especially under
high ALWC conditions. Fig. 5b and d showed that $f_{29}$ was highly correlated with
aqSOA formation during CP and PP. Moreover, it should be noted that a strong
anticorrelation between aqSOA and OOA concentrations was observed during PP at
ALWC > 80 μg m$^{-3}$ when $f_{29}$ > 0.15 ($p$ < 0.001), but not during CP. These results
indicated that considerable aqSOA might be formed from BBOA, which was more
intensive than OOA at high ALWC levels during PP.

Previous research demonstrated that $f_{44}$ could be used as a tracer of aged SOA,

$f_{43}$ (normalized mass spectrum signal at m/z 43) as a tracer of POA and fresh SOA,
and $f_{60}$ as a tracer of BBOA (Cubison et al., 2011; Ng et al., 2010). The triangle plots
of $f_{44}$ versus $f_{43}$ and $f_{44}$ versus $f_{60}$ have been widely used to characterize OA evolution,
the ratio changes of $f_{44}$ versus $f_{43}$ and $f_{44}$ versus $f_{60}$ as the functions of atmospheric and
BBOA aging, respectively (Ortega et al., 2013; Paglione et al., 2020; Xu et al., 2017;
Xu et al., 2019). As shown in Fig. 6, the bottom region of the triangle was dominated
by BBOA, CCOA, and HOA with low $f_{44}$ (0.040, 0.017, and 0.016, respectively) in
this study. However, the $f_{44}$ of SOA factors (i.e., OOA and aqSOA) (0.108 and 0.117)
were observably higher than POA factors, showing the freshly oxidized properties of
SOA and further aging of OOA might also form aqSOA. Meanwhile, $f_{44}$ of aqSOA
was close to that observed in fogs (Gilardoni et al., 2016; Kim et al., 2019),
highlighting the presence of aqueous-phase reactions in this study. Fig. 4c showed
BBOA and aqSOA with much higher $f_{60}$ values (0.019 and 0.011) than CCOA (0.009)
and HOA (0.008). The $f_{60}$ value of OOA was 0.002 lower than the typical background



value (0.003) in the atmospheric without biomass burning influence (Cubison et al.,
2011). The mass spectrometry feature of aqSOA showed large $f_{44}$ (representation of
aged OA) and $f_{60}$ (presence of anhydrosugars) values, laying in a schematic space of
aged BBOA based on mass spectrometry features in previous research (Cubison et al.,
2011; Ortega et al., 2013). This suggested that BBOA could be the important
precursors for aqSOA instead of OOA via aqueous-phase reactions. These results
were consistent with previous research and most of the observation data were within
the triangle space, indicating that POA factors were freshly emitted and aqSOA were
more oxidized from aged BBOA.

During PP, the $f_{44}$ values ranging from 0.022 to 0.140 (0.080 ± 0.035) were

significantly higher than that during CP (0.021−0.150, 0.064 ± 0.019) ($p < 0.001$),
while the $f_{43}$ value was slightly lower with an average of 0.062 ± 0.027. Compared
with CP (0.17 and −0.53), $f_{44}$ showed a more significant increase as the decreasing of
$f_{43}$ with higher $r^2$ value (0.70) and the regression slope of $f_{44}$ versus $f_{43}$ (−1.09) was
close to −1 during PP. This indicated that more aged SOA existed in the atmosphere
during PP (Fig. 6a and c). It should be noted that the points of $f_{44}$ versus $f_{43}$ were
inside the upper boundary of the triangle region, even some were outside the bottom
boundary of the triangle region during CP and PP, suggesting that the formation of
less oxidized SOA via aqueous-phase reactions instead of photo-chemical reactions
(Kim et al., 2019; Zhao et al., 2019). Moreover, these points outside the bottom
boundary of the triangle region with higher $f_{44}$ (> 0.05) and lower $f_{43}$ (< 0.06) showed
relatively higher ALWC during PP, but not during CP. Overall, these results



highlighted the aqSOA with less oxidized formation via aqueous-phase reactions
during PP.

Here, the triangle plots of $f_{44}$ versus $f_{60}$ colored by ALWC under different $PM_{2.5}$

pollution levels were analyzed (Fig. 6b and d), when the link between aqSOA and
BBOA was further stressed by a schematic representation of aged BBOA. The
contribution of $f_{60}$ to different OA factors in this study and previous research was
represented in Fig. 6b and d (Bao et al., 2023; Gilardoni et al., 2016; Kim et al., 2019;
Ng et al., 2011; Paglione et al., 2020; Xu et al., 2015; Xu et al., 2017; Xu et al., 2019;
Zhao et al., 2017; Zhao et al., 2019). The background space ($f_{60} < 0.003$) without
biomass burning influence was also shown by the grey shaded area. All the $f_{60}$ values
were higher than 0.003 and most points fell in the triangular region, suggesting the
contribution of biomass burning to OA. During PP, The $f_{60}$ values ranging from 0.005
to 0.019 (0.010 ± 0.004) were similar with CP (0.004−0.019, 0.010 ± 0.003), while
the mean $f_{44}$ value was significantly higher. Compared with CP, most of the data
points with high ALWC fell in the schematic space of aged BBOA, and a stronger
negative correlation between $f_{44}$ and $f_{60}$ was observed ($r^2 = 0.72$, $p < 0.001$) during PP.
Overall, these results pointed to the fact that the observed aqSOA was originated from
the aged BBOA via aqueous-phase reactions under high ALWC during PP.



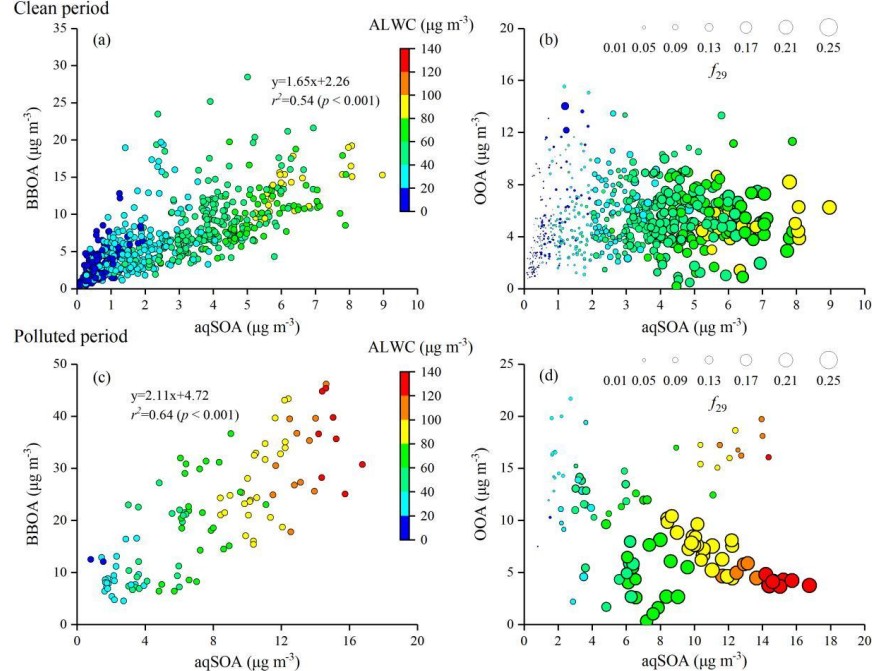


**Figure 5.** Scatter plots of aqSOA versus **(a, b)** BBOA and **(c, d)** OOA mass concentrations

410 colored by ALWC during clean period and polluted period. The size of the symbols in **(b)** and **(d)**

411 increases with the increase of the $f_{29}$ value, which is a tracer for alcohol compounds.

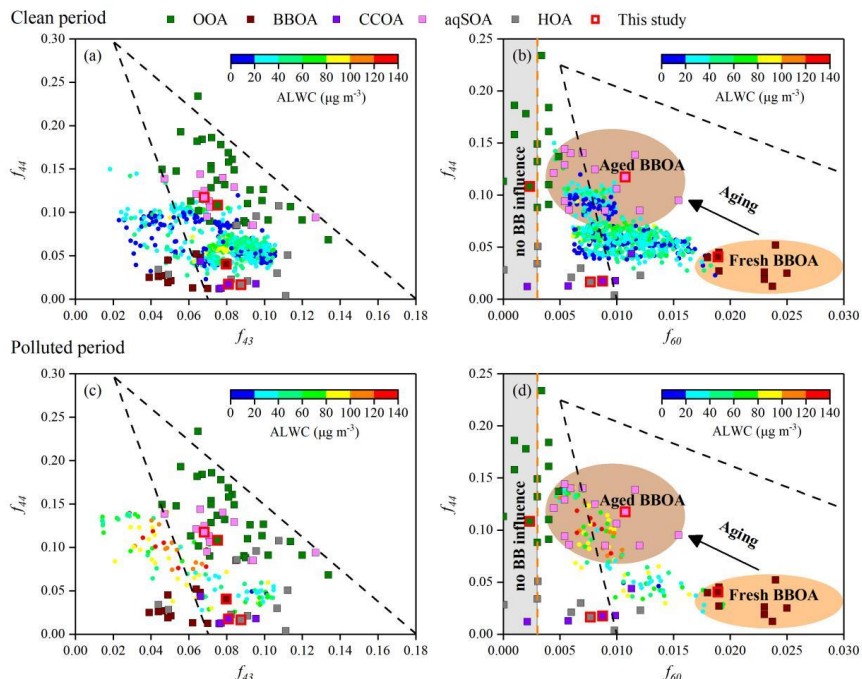

**Figure 6.** Triangle plots of **(a, c)** $f_{44}$ (normalized mass spectrum signal at m/z 44) versus $f_{43}$

(normalized mass spectrum signal at m/z 43), and **(b, d)** $f_{44}$ versus $f_{60}$ (normalized mass spectrum

signal at m/z 60) colored by ALWC during clean period and polluted period. The dashed lines in

**(a)** and **(c)** were derived from Ng et al. (2010) and used to follow the aging of OA components in

the atmosphere. The background value of secondary aged OA (brown dashed line) and the black

dashed lines characterising the aging of BBOA in **(b)** and **(d)** were derived from Cubison et al.

(2011). The data points included the measurements in previous studies (Bao et al., 2023; Gilardoni

et al., 2016; Kim et al., 2019; Ng et al., 2011; Paglione et al., 2020; Xu et al., 2015; Xu et al., 2017;

Xu et al., 2019; Zhao et al., 2017; Zhao et al., 2019). $f_{43}$ (mainly $C_2H_3O^+$) is a tracer for POA and

fresh SOA. $f_{44}$ is a proxy of the OA oxygenation degree and used as a tracer for aged SOA. $f_{60}$ is a

proxy of anhydrosugars emitted from biomass burning.



**3.3 Evolution of BrC Absorption**
Considerable OA from fresh and aged biomass-burning emissions exhibited
absorption properties across UV to Vis range with much higher AAE value than BC
(Laskin et al., 2015), might contribute to a net positive radiative forcing. Therefore,
the BrC absorption properties and their relationship with five OA factors were
analyzed in this study. The values of $Abs_{\lambda,BrC,pri}$ and $Abs_{\lambda,BrC,sec}$ were obtained by MRS
method, and the MLR method was used to estimate Abs of five OA factors at each
wavelength (SI Text S2,S3). As shown in Fig. S6, the average value of $Abs_{370,BrC}$ was
$42.4 \pm 28.5$ Mm$^{-1}$ (accounting for 49.2% of $Abs_{370}$), much higher than $Abs_{660,BrC}$ (2.6
$\pm 1.3$ Mm$^{-1}$, 10.5%), suggesting a high absorption efficiency for BrC in the near-UV
wavelength. The $Abs_{\lambda,BrC,pri}$ and $Abs_{\lambda,BrC,sec}$ accounted for 56.8%−72.5% and
27.5%−43.2% of $Abs_{\lambda,BrC}$ from 370 nm to 660 nm respectively, indicating POA was
the dominant OA component affecting BrC absorption. However, the contribution of
$Abs_{\lambda,BrC,sec}$ to $Abs_{\lambda,BrC}$ increased with wavelength, suggesting the impact on $Abs_{BrC}$
from SOA should not be ignored. Here we showed that aqSOA formation from aged
BBOA contributed to the BrC budget and was strong absorption across UV to Vis
range.
The data at 370 nm with higher signal-to-noise ratios and $Abs_{BrC}$ contribution
was chosen to further analyze the correlations of BrC absorption with various OA
components. Compared with CCOA, HOA, and OOA (12.3%, 9.1%, and 11.1%), the
Abs at 370 nm calculated for BBOA ($Abs_{370,BBOA}$) and aqSOA ($Abs_{370,aqSOA}$) showed
higher contributions (51.9% and 16.4%) to $Abs_{370,BrC}$, consistent with the higher MAC




values (Fig. 7). Fig. S8 presented the correlations between $\text{Abs}_{370,\text{BrC}}$ and the mass
concentrations of OOA, BBOA, CCOOA, aqSOA, HOA, and $\text{C}_2\text{H}_4\text{O}_2^+$. $\text{Abs}_{370,\text{BrC}}$
showed the strongest positive correlations with BBOA and $\text{C}_2\text{H}_4\text{O}_2^+$ (ion fragments
tracers of BBOA) concentrations ($r^2 = 0.77$, $p < 0.001$), followed by aqSOA
concentrations ($r^2 = 0.69$, $p < 0.001$). In contrast, the correlations with HOA
concentrations ($r^2 = 0.36$), CCOA ($r^2 = 0.25$), and OOA ($r^2 = 0.09$, $p > 0.1$) were
much weaker. The high value and contribution of $\text{Abs}_{370,\text{aqSOA}}$ and strong positive
correlation between $\text{Abs}_{370,\text{BrC}}$ and aqSOA concentrations could be likely that a
portion of aqSOA was formed from aged BBOA via aqueous-phase reactions.
Gilardoni et al. (2016) demonstrated that aqSOA formation from aged BBOA via
aqueous-phase reactions in the ambient atmosphere contributed to the BrC budget and
exhibited slightly higher $\text{AAE}_{467–660}$ (AAE of aerosols from 467 nm to 660 nm )
values than the fresh and processed biomass-burning emissions in laboratory
experiments. The MAC values of the five resolved OA components were shown in
Fig. 7. Among these, BBOA showed the highest MAC value (2.37 m$^2$ g$^{-1}$), followed
by aqSOA (1.23 m$^2$ g$^{-1}$) at 370 nm, indicating that the oxidation of BBOA to aqSOA
decreased light absorption at short wavelengths. Previous research found that the
MAC of BBOA was twice that of SOA associated with water-soluble BrC, such as
MAC of BBOA (1.3 ± 0.06 m$^2$ g$^{-1}$) was much higher than that for other OA factors at
365 nm (Lorenzo et al., 2017; Washenfelder et al., 2015). However, it should be noted
that aqSOA had the lowest $\text{AAE}_{370–660,\text{aqSOA}}$ value (3.54), while BBOA has the highest
$\text{AAE}_{370–660,\text{BBOA}}$ value (4.93). Moreover, the contribution of aqSOA to $\text{Abs}_{\text{BrC}}$





increased from 16.4% to 26.7% from 370 to 660 nm, while the contribution from
BBOA decreased from 51.9% to 39.1% from 370 to 660 nm. These suggested aqSOA
formation from aged BBOA might play an important role in the light absorption of
BrC across UV to Vis range.

Fig. S9 showed the ternary contour map to quantify the contribution of BBOA,

CCOA, and HOA factors to $Abs_{370,BrC,pri}$, when the strong positive correlation ($p <$
0.001) and high slope of the linear regression (1.80) between BBOA mass
concentration and $Abs_{370,BrC,pri}$ were observed. Among these POA factors, the high
mass fractions of BBOA to POA were consistent with the high $Abs_{370,BrC,pri}$ values
(Fig. S9a). For example, the most data of $Abs_{370,BrC,pri}$ higher than 49.1 Mm$^{-1}$ (90th
percentile of $Abs_{370,BrC}$) fell in the region of high BBOA/POA values (> 0.5).
Moreover, $Abs_{370,BrC,pri}$ significantly increased with the increases of BBOA and
$C_2H_4O_2^+$ mass concentrations with higher $r^2$ values (0.63 and 0.55) than HOA and
CCOA (0.19 and 0.14) (Fig. S9b). These results indicated the major contribution of
BBOA to primary BrC light absorption.

During the campaign, the relationship between $Abs_{370,BrC,sec}$ and SOA factors

mass concentrations was analyzed to understand the correlation between secondary
BrC absorption and its chromophores. As shown in Fig. S10, $Abs_{370,BrC,sec}$ significantly
increased with the increase of aqSOA concentrations ($r^2 = 0.44$, $p < 0.001$) and high
$Abs_{370,BrC,sec}$ values were consistent with the high ALWC values, but not OOA ($p >$
0.1). Higher slope of the linear regression (3.50) at 370 nm and MAC values across
UV to Vis range of aqSOA were also observed than OOA (Fig. 7). To further



characterize the evolution of secondary BrC absorption, $Abs_{370,BrC,sec}$ was normalized
by $\Delta CO$ (the background-corrected CO mixing ratios) to minimize the effect of
boundary layer height (Fig. 8) (DeCarlo et al., 2010). Here the background CO value
(400 ppb) was defined as the lowest 1.25th percentile of the CO values during the
campaign (Kondo et al., 2006). Fig. 8 showed that the values of $Abs_{370,BrC,sec}/\Delta CO$
increased with the increases of aqSOA and ALWC concentrations especially at night
(from 17:00 to 03:00 LT), while $Abs_{370,BrC,pri}/\Delta CO$ decreased with the increases of
BBOA and $C_2H_4O_2^+$ concentrations at night. This suggested considerable secondary
BrC chromophores with strong absorption at 370 nm were formed under the high
ALWC at night, which might be related to the aqSOA from the aged BBOA via
aqueous-phase reactions (Pang et al., 2019; Yu et al., 2016; Zhao et al., 2014). The
low values of $Abs_{370,BrC,sec}/\Delta CO$ at 12:00−14:00 LT should be related to the photolysis
and/or photooxidation causing BrC photobleaching (Sareen et al., 2013; Zhao et al.,
2015). Overall, we suggested that aqSOA formed from biomass-burning emissions
might be important for BrC absorption, especially at night.

$AAE_{370–880}$ was another key parameter to characterize the absorption properties

of aerosols, its correlations with the mass fraction of aqSOA ($f_{aqSOA}$) and BBOA
($f_{BBOA}$) to OA, and BC-to-OA ratios were shown in Fig. 9. During the campaign, the
strong positive correlation ($r^2 = 0.49$, $p < 0.001$) between $AAE_{370–880}$ and $f_{aqSOA}$ was
observed with $AAE_{370–880}$ values up to 2.65, while $AAE_{370–880}$ values increased with
the slight increase of $f_{BBOA}$ in general ($r^2 = 0.21$, $p < 0.001$) (Fig. 9a and c). Previous
laboratory research indicated that the biomass-burning emissions influence on the Abs





could be reflected in the relationship between AAE and BC-to-OA ratios (Lu et al.,
2015; Saleh et al., 2014). Fig. 9b showed this relationship and $AAE_{370-880}$ values were
successfully parameterized by BC-to-OA ratios during the campaign ($r^2 = 0.51$, $p <$
0.001). The parameterized curve (black curve) and these data points measured in this
study were similar to those reported in the previous laboratory research on
biomass-burning emissions using the wavelength from 370 nm to 950 nm (red curve)
(Lu et al., 2015). Here, 950 nm and 880 nm were used as the highest wavelength
respectively, and similar values were found between $AAE_{370-950}$ and $AAE_{370-880}$
(within 10.0%). It should be noted that the data points of high $AAE_{370-880}$ were
consistent with the low BC-to-OA ratios and large $f_{aqSOA}$ values in general. Moreover,
the average value of $AAE_{370-880}$ observed in this study (1.95) was higher than
$AAE_{370-950}$ observed in the laboratory experiments of fresh and photo-chemically
aged biomass-burning emissions (i.e., 1.38 and 1.48 for fresh oak and pocosin pine,
1.42 and 1.73 for aged oak and pocosin pine) (Saleh et al., 2013).

Overall, our ambient observations highlighted the importance of aqSOA

formation from aged biomass-burning emissions in contributing to the BrC budget
and light absorption, reinforcing aqSOA was an important role in the Sichuan Basin
and should be accounted in the air quality budget and climate forcing balance.



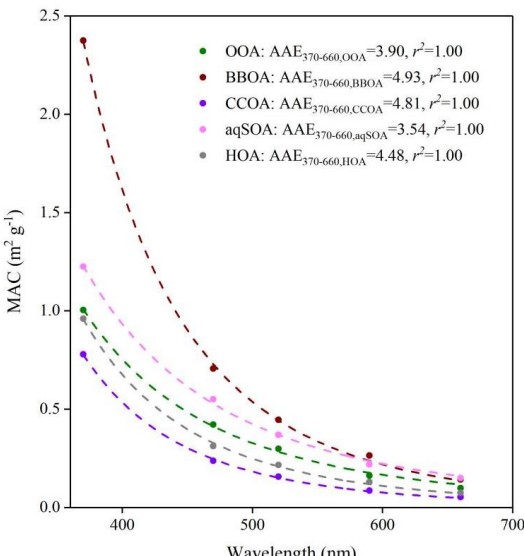


**Figure 7.** MAC of different OA factors as a function of wavelength from 370 to 660 nm.

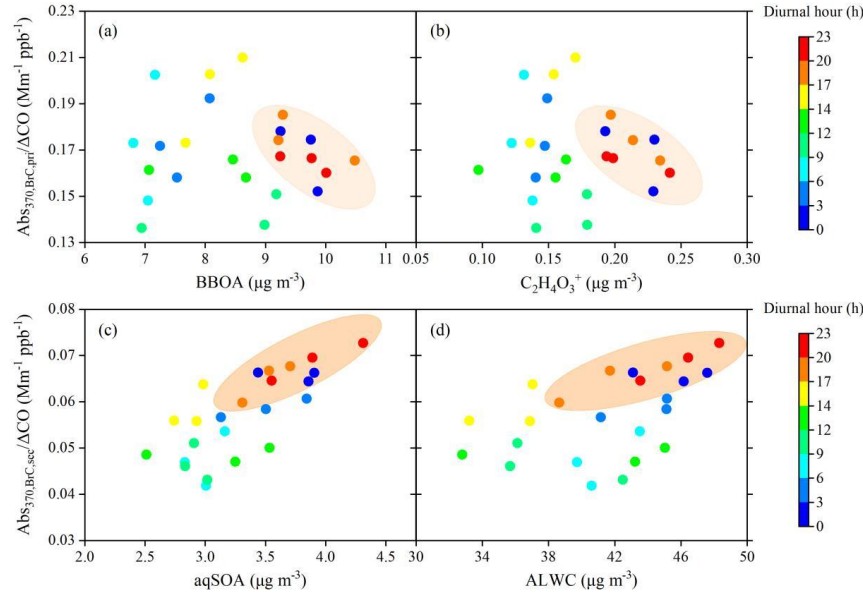


**Figure 8.** Scatter plots of $Abs_{370,BrC,pri}/\Delta CO$ versus **(a, b)** BBOA and $C_2H_4O_2^+$ mass concentrations

and $Abs_{370,BrC,sec}/\Delta CO$ versus **(c, d)** aqSOA and ALWC colored by the local time.



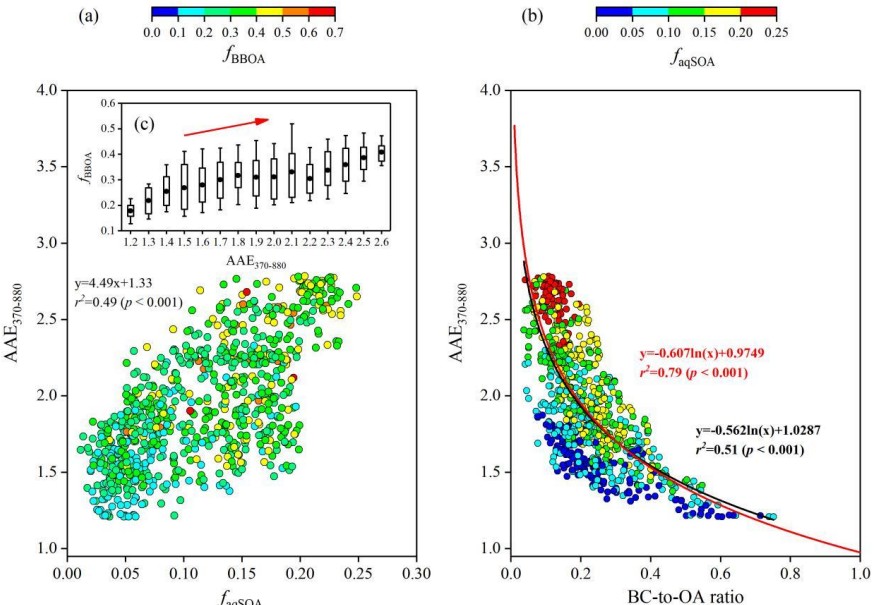

**Figure 9.** Relationship between **(a)** $AAE_{370-880}$ and the mass fraction of aqSOA ($f_{aqSOA}$ = aqSOA/OA) colored by the mass fraction of BBOA ($f_{BBOA}$ = BBOA/OA), and **(b)** BC-to-OA ratios colored by $f_{aqSOA}$. **(c)** Variations of $f_{BBOA}$ as a function of $AAE_{370-880}$. The red curve in **(b)** was the best fit curve to data taken from Lu et al. (2015) and described the Abs of fresh and aged BBOA.

## 4 Implications

Organic aerosol (OA) was the dominant component of atmospheric aerosol with significantly implications for air quality and climate forcing. Field observations indicated that secondary organic aerosol (SOA) accounted for most of OA worldwide and aqueous-phase oxidation was an important pathway for the SOA formation. An increasing laboratory research demonstrated that the aqueous secondary organic aerosol (aqSOA) formed from biomass-burning emissions via aqueous-phase



reactions could lead to positive radiative forcing and influence atmospheric
photochemistry indirectly (Drozd and McNeill, 2014; Herrmann et al., 2015; Kampf
et al., 2012; Nozière and Esteve, 2007; Ye et al., 2019). Our results revealed the
aqSOA formation and brownness from aged BBOA via aqueous-phase reactions by
the direct ambient observation and highlighted the importance of aqSOA on air
quality and climate. The aqSOA formation from aged BBOA contributed to the brown
carbon (BrC) budget and showed stronger absorption across ultraviolet to visible
range than other OA components. Therefore, the aqueous-phase oxidation of
biomass-burning emissions should be taken into account in air quality and climate
models for a correct description of the global OA budget and its climate-relevant
optical properties.
In this work, the aqSOA formation and absorption properties in the ambient
atmosphere were observed directly in the Sichuan Basin, China. The results
demonstrated the fact that considerable aqSOA was originated from the aged
biomass-burning emissions via aqueous-phase reactions under high ALWC in the
ambient atmosphere. Additionally, the less oxidized aqSOA formation via
aqueous-phase reactions instead of photo-chemical reactions played a key role in the
haze pollution dynamic evolution during the polluted period (Figure 6). This study
also indicated that the impact on secondary BrC absorption should not be ignored,
although primary BrC dominated the BrC absorption across ultraviolet to visible
range. The results of this study highlighted the importance of aqSOA formation from
aged biomass-burning emissions in contributing to the BrC budget and absorption,



especially at night. Figure 9 further showed that the similarity between ambient data
and the parameterized curve of $AAE_{370-880}$ versus BC-to-OA ratios was consistent
with the laboratory research on biomass-burning emissions. Higher values of
$AAE_{370-880}$ and $MAC_{\lambda,aqSOA}$ reinforced the stronger absorption of aqSOA formation
from aged biomass-burning emissions via aqueous-phase reactions than
photo-chemically reactions.
In conclusion, our ambient observation demonstrated that the formation and
brownness of aqSOA from the aged biomass-burning emissions in the Sichuan Basin,
China. Brown aqSOA originating from biomass-burning emissions was an important
and unaccounted player in air quality budget and climate forcing balance worldwide.
This study was helpful in understanding the formation, light properties, and impacts
of aqSOA in the ambient atmosphere. Future research should focus on the
molecular-level characterization, transportation, and reactivities of gas and
particle-phase aqSOA precursors to improve understanding of aqSOA formation
processes and absorption properties.

**Data availability.** The data generated and analysed in this study are available from
https://doi.org/10.5281/zenodo.14626304 (Peng et al., 2025).

**Author contributions.** CZ, CP, YD, and ZL designed the experiments. Data analysis
and interpretation were performed by CP, ZT, HT, KZ, ZL, and GS. CP, XY, and MT
wrote the paper. ZT, YC, XL, LZ, YC, and YF contributed to the paper with useful



scientific discussions or comments.

**Competing interests.** The authors declare that they have no conflict of interest.

**Acknowledgements.** This study was supported by the Natural Science Foundation of
Chongqing Municipality (No. CSTB2022NSCQ-MSX0504), National Natural
Science Foundation of China (No. 42305126), National Key Research and
Development Program of China (No. 2023YFC3709301).



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
