# Peer review of "Measurement report: Formation and brownness of aqueous secondary organic aerosol from the aged biomass-burning emissions in the Sichuan Basin, China"

_EGUsphere, 2025_

## Author Comment (AC1)

Dear Reviewers,

We appreciate the comments of reviewers and have revised the manuscript to address the review comments from all reviewers. We have provided a point-to-point response to the concerns, suggestions, and questions. We hope our work could satisfy both the reviewers and the editor. Our answers to each question have been marked in blue text. The annotated line numbers refer to the revised copy of the manuscript.

This study reported the direct ambient observation of SOA formation and absorption properties in the aqueous phase from the Sichuan Basin, China. The results showed considerable aqSOA was originated from the aged biomass-burning emissions via aqueous-phase reactions instead of photo-chemical reactions under high aerosol liquid water content (ALWC) conditions, especially during the polluted period. This study revealed the aqSOA formation and brownness from aged biomass-burning emissions and highlighted the importance of aqueous-phase reactions on air quality and climate. This is a manuscript with potential. I recommend the manuscript be revised before being accepted for publication.

1. Lines 83-87. The authors mention the geographic variability of the North China Plain amount the Sichuan Basin, but there is a study on aqSOA in these two regions is not described in detail, it is recommended to add a comparative analysis to better highlight the regional uniqueness of this study.

**Response:** Thanks for the reviewer's comment. It has been revised accordingly in the revised manuscript, which reads "However, observations on aqSOA formation and

optical properties in China are limited and most research concentrate on the North China Plain (NCP). Similar to NCP, the Sichuan Basin (SCB) characterized by high humidity and frequent biomass burning is also the main region with severe aerosol pollution in China (Tian et al., 2019; Wang et al., 2018; Yang et al., 2011). Previous research indicated that aqSOA from different regions exhibited distinct formation mechanisms and optical properties, due to the diverse sources and ambient conditions (Bao et al., 2023; Bao et al., 2024; Wang et al., 2021; Xu et al., 2017). For instance, Wang et al. (2021) revealed fast aqueous-phase conversion of fossil-fuel primary organic aerosol (FF-POA) to aqSOA under high-humidity conditions during a Beijing winter haze event, and found that aqSOA exhibited much lower light absorption than its primary precursor due to decreased aromaticity. Similarly, Huang et al. (2023) illustrated that the aqueous-phase oxidation of fossil fuel combustion emissions played a critical role in SOA formation under high RH conditions. Unlike these studies in NCP, the effect of aqueous-phase reactions on oxygenated OA (OOA) formation was significant when aerosol liquid water content (ALWC) was below 200 $\mu g\ m^{-3}$, but was insignificant when ALWC > 200 $\mu g\ m^{-3}$ in SCB. Additionally, the aqueous-phase oxidation process probably did not play a role in the decay of BrC during summer in Chengdu (Bao et al., 2024). Currently, few studies explored the dynamic evolution and optical properties of aqSOA, and the knowledge of ambient aqSOA processing is still limited in SCB. Therefore, a more detailed characterization of aqSOA formation and optical properties is of great importance to reveal the key factors contributing to haze formation." **(Lines 82-104).**

2. Section 2.1. As a field measurement report, this paper lacks a map of the geographic location of the observatory sites, which does not allow us to visualize the

geographic information about the Sichuan Basin, so we suggest that a map of the observatory sites and the surrounding terrain be added.

**Response:** A map of the observatory site and the surrounding terrain has be added in the revised SI:

[Figure]

**Figure S1.** Location of the observation site in the Sichuan Basin (from Baidu Maps, © 2025 Baidu – GS(2023)3206).

3. Line 151. Further details are required on the basis for setting the CE value of ToF-ACSM to 0.5.

**Response:** Thanks for the reviewer's comment. It has been revised accordingly in the revised manuscript, which reads "Meanwhile, a particle collection efficiency (CE) was introduced to compensate for the particle loss, as the acidity, the contribution of ammonium nitrate (ANMF) and phase state changed the particle bounce effects at the vaporiser (Matthew et al., 2008). Middlebrook et al. (2012) developed a CE algorithm for ToF-ACSM to quantify the aerosol species. Their results indicated that a constant CE value of 0.45 should be used when: (1) the ANMF is below 40%, or (2) particles

are partially or fully neutralized. In this study, aerosol particles were dried by Nafion dryer (RH < 30%) before sampling by ToF-ACSM, and the ANMF was always below 40%. As shown in Fig. S2, the average ratio of the measured $NH_4$ to the predicted $NH_4$ needed to fully neutralize the $SO_4$, $NO_3$ and Chl was approximately 1. All of these conditions did not affect the CE value that had usually been used at this site. The typical default CE value (0.5) was applied during the whole sampling period, which was consistent with previous research (Bao et al., 2025; Peng et al., 2025; Sun et al., 2016a; Sun et al., 2016b; Zhao et al., 2019). While the typical default CE is 10% higher than 0.45, the difference is small considering the 30% uncertainty determined for CE (Bahreini et al., 2009). Additionally, the strong correlation between NR-PM$_{2.5}$ and PM$_{2.5}$ mass concentrations supported that the CE value was reasonable (Fig. S3)." **(Lines 165-182).**

4. Section 2.3.2. The ISORROPIA-II model estimates ALWC from these inorganic constituents only, but will the organic constituents affect the estimates, especially at higher OA conditions.

**Response:** Thanks for the reviewer's comment. In this study, the organic contribution for ALWC was calculated by Zdanovskii−Stokes−Robinson (ZSR) mixing rule as discussed in SI Text S2 (Guo et al., 2015; Huang et al., 2020; Nguyen et al., 2016; Xu et al., 2022). The average ALWC with organic species only took up $3.7 \pm 2.2\%$ of total ALWC, thus the ALWC was determined only considering inorganic species in this study. It has been revised accordingly in the revised manuscript, which reads

**"2.3.2 Aerosol liquid water content**

The ALWC is controlled by meteorological conditions (T and RH) and also by inorganic and organic components. During the campaign, the ALWC with inorganic

species was estimated by the ISORROPIA-II model based on the ammonium, nitrates, sulfates, and chlorides mass concentrations from ToF-ACSM and the meteorological parameters (T and RH) from National Environmental Monitoring Station (Fountoukis and Nenes, 2007). Here, the forward type and metastable mode were used in the ISORROPIA-II model (Hennigan et al., 2015). The thermodynamic equilibrium of the $NH_4^+-SO_4^{2-}-NO_3^--Cl^--H_2O$ system was modeled and ALWC was then calculated. Consistent with previous research, the organic contribution for ALWC was calculated by Zdanovskii−Stokes−Robinson (ZSR) mixing rule as discussed in SI Text S2 (Guo et al., 2015; Huang et al., 2020; Nguyen et al., 2016; Xu et al., 2022). In this study, the ALWC with organic species ranged from 0.1 to 35.2 µg m$^{-3}$, with an average of 1.9 ± 3.0 µg m$^{-3}$, taking up 3.7 ± 2.2% of total ALWC. As organic species had minor effects on total ALWC (< 5%), the ALWC was determined only considering inorganic species (Chen et al., 2021; Guo et al., 2015; Liu et al., 2017)." **(Lines 197-212).**

"**Text S2.** Estimation of Aerosol Liquid Water Content (ALWC)

The effect of inorganic ions on ALWC was estimated using a thermodynamic equilibrium model for the $NH_4^+-SO_4^{2-}-NO_3^--Cl^--H_2O$ aerosol system, ISORROPIA-II (Fountoukis and Nenes, 2007). Here we ran the ISORROPIA II in forward mode, and the particles were assumed to be deliquescent, i.e., in metastable mode (Hennigan et al., 2015). The ammonium, nitrates, sulfates, and chlorides mass concentrations were measured by ToF-ACSM and the meteorological parameters (T and RH) were obtained from National Environmental Monitoring Station.

Meanwhile, the contribution of organics to ALWC was calculated by Zdanovskii−Stokes−Robinson (ZSR) mixing rule using the following equation (Guo et al., 2015; Huang et al., 2020; Nguyen et al., 2016; Xu et al., 2022):

$$W_{OA} = \frac{OA}{\rho_{OA}} \rho_W \frac{k_{OA}}{(100\%/a_w - 1)} \tag{S4}$$

where OA was the mass concentration of organics, $\rho_w$ was the density of water (1.0 g cm$^{-3}$) and $\rho_{OA}$ was the density of organics (1.4 g cm$^{-3}$) (Nguyen et al., 2016; Xu et al., 2022). $k_{OA}$ was the hygroscopicity parameter of organic aerosol composition. We applied $k_{OA}$ value of 0.08 for organic aerosols at urban site (Nguyen et al., 2016). $a_w$ was the water activity (dimensionless). We assumed $a_w$ was equivalent to RH for the sake of simplicity because of a lack of particle diameter data (Huang et al., 2020; Xu et al., 2022)."

5. Lines 220-222. There is a need for further clarification on the judgment of clean-up and contamination periods, the standards are different in each country and it is recommended to add comparisons with other standards.

**Response:** Thanks for the reviewer's comment. It has been revised accordingly in the revised manuscript, which reads "According to the Chinese National Ambient Air Quality Standard (NAAQS) (GB 3095-2012) (MEP, 2012), the Grade I annd Grade II levels for daily PM$_{2.5}$ mass concentration are 35 µg m$^{-3}$ and 75 µg m$^{-3}$, respectively. The Chinese NAAQS Grade II level, based on WHO's Phase-1 interim target (IT-1), is higher than the WHO Air Quality Guideline (AQG) value (15 µg m$^{-3}$), the EU daily limit (25 µg m$^{-3}$), and U.S. 24-hour standard (35 µg m$^{-3}$). During the campaign, the average of PM$_{2.5}$ mass concentration was 1.4 times NAAQS Grade I level (35 µg m$^{-3}$). Therefore, the pollution periods (PP) were defined by the daily PM$_{2.5}$ mass

concentration exceeding NAAQS Grade II level of 75 µg m$^{-3}$. Similarly, the days with PM$_{2.5}$ mass concentration below 75 µg m$^{-3}$ were characterized as clean periods (CP). " **(Lines 261-270).**

6. Lines 294-296. The authors suggest that aqSOA may also be produced by aqueous phase reactions of fossil fuel-related OA components, but this is only speculative and a more convincing basis is needed.

**Response:** We agree that the suggestion regarding aqSOA production from fossil fuel-related OA components is speculative. Our current study does not provide direct experimental evidence to definitively support this hypothesis and can not specify the physical or chemical mechanisms behind the fossil fuel-related OA to aqSOA transformation. However, the observed correlation between fossil-fuel-related OA components and ALWC is consistent with recent research (Wang et al., 2021). Moreover, previous studies indicated that aqueous-phase processes could play a role in the formation of SOA from fossil fuel emissions (Ervens et al., 2011; Huang et al., 2023; Wang et al., 2021; Xu et al., 2022; Yan et al., 2017). For example, Wang et al. (2021) and Xu et al. (2022) have highlighted the potential for aqueous-phase reactions to contribute to SOA formation, particularly in regions with high levels of anthropogenic emissions. Based on these findings, we suggest that while our data do not conclusively prove this mechanism, further investigation is necessary to strengthen this hypothesis and explore its relevance in different atmospheric environments.

It has been revised accordingly in the revised manuscript, which reads "Additionally, previous studies indicated that aqueous-phase processes could play a role in the formation of SOA from fossil fuel emissions (Ervens et al., 2011; Huang et al., 2023;

Wang et al., 2021; Xu et al., 2022; Yan et al., 2017). For example, Wang et al. (2021) and Xu et al. (2022) have highlighted the potential for aqueous-phase reactions to contribute to SOA formation, particularly in regions with high levels of anthropogenic emissions. In this study, a strong anticorrelation between the mass fraction of fossil-fuel related OA components (sum of CCOA, HOA and OOA) and ALWC at the high $f_{29}$ values was also observed ($r^2 = 0.48$, $p < 0.001$) (not shown), consistent with recent research (Wang et al., 2021). This indicated that aqSOA might also be produced by aqueous-phase reactions of fossil-fuel related OA components." **(Lines 365-375).**

7. Lines 350-352. These results indicated that considerable aqSOA might be formed from BBOA, which was more intensive than OOA at high ALWC levels during PP. What are the authors' judgments about the magnitude of their intensity, and what are the differences in "intensive"?

**Response:** Thanks for the reviewer's comment. We have revised this sentence and provided a detailed discussion on the correlations between ALWC and both aqSOA and OOA under high ALWC conditions during PP and CP to clarify the differences in "intensive". It has been revised accordingly in the revised manuscript, which reads "Moreover, the mass fraction of aqSOA showed a stable increasing trend and remained high levels (from 18% to 22%) at ALWC > 80 μg m$^{-3}$, which was associated with a corresponding decrease in OOA (from 15% to 10%) (Fig. 4b). Compared with OOA ($p > 0.1$), the aqSOA concentrations showed strong positive correlation with ALWC ($r^2 = 0.73$, $p < 0.001$) when ALWC > 80 μg m$^{-3}$ during PP. In contrast, ALWC showed the weak correlations with aqSOA and OOA concentrations during CP ($p > 0.1$). It should be noted that a strong anticorrelation between aqSOA and OOA concentrations was observed during PP at ALWC > 80 μg m$^{-3}$ when $f_{29}$ > 0.15 ($r^2 = 0.76$, $p < 0.001$), but not during CP ($p > 0.1$) (Fig. 5b and d). These results

indicated that the aqSOA formation was more intensive than OOA at high ALWC levels during PP." **(Lines 431-441).**

8. Lines 372-373. These results indicated that considerable aqSOA might be formed from BBOA, which was more intensive than OOA at high ALWC levels during PP. Any details on what the mechanism is for this result?

**Response:** Thanks for the reviewer's comment. It has been revised accordingly in the revised manuscript, which reads "Moreover, the mass fraction of aqSOA showed a stable increasing trend and remained high levels (from 18% to 22%) at ALWC > 80 $\mu$g m$^{-3}$, which was associated with a corresponding decrease in OOA (from 15% to 10%) (Fig. 4b). Compared with OOA ($p > 0.1$), the aqSOA concentrations showed strong positive correlation with ALWC ($r^2 = 0.73$, $p < 0.001$) when ALWC > 80 $\mu$g m$^{-3}$ during PP. In contrast, ALWC showed the weak correlations with aqSOA and OOA concentrations during CP ($p > 0.1$). It should be noted that a strong anticorrelation between aqSOA and OOA concentrations was observed during PP at ALWC > 80 $\mu$g m$^{-3}$ when $f_{29} > 0.15$ ($r^2 = 0.76$, $p < 0.001$), but not during CP ($p > 0.1$) (Fig. 5b and d). These results indicated that the aqSOA formation was more intensive than OOA at high ALWC levels during PP." **(Lines 431-441).**

Additionally, we have shown the details on the mechanism for this result: "The relative abundance of m/z 45 (mainly $HCO_2^+$), a tracer ion for carboxylic acids, was higher in the aqSOA spectra than in the OOA spectrum (Fig. S9). It was consist with previous research which found that aqueous-phase reactions were important sources of oxygenated organic compounds, including organic acids (Ervens et al., 2011; Kim et al., 2019; McNeill, 2015; Sun et al., 2010; Yu et al., 2014). Fig. 6b shows BBOA and aqSOA with higher $f_{60}$ values (0.019 and 0.011) than CCOA (0.009) and HOA

(0.008). The $f_{60}$ value of OOA was 0.002, lower than the typical background value (0.003) in the atmospheric without biomass burning influence (Cubison et al., 2011). The mass spectrometry feature of aqSOA showed large $f_{44}$ and $f_{60}$ values, laying in a schematic space of aged BBOA based on mass spectrometry features in previous research (Cubison et al., 2011; Ortega et al., 2013). Additionally, BBOA contains abundant water-soluble organic compounds (WSOC) with (i.e., sugars, phenols, and organic acids), that can form aqSOA via efficient aqueous-phase reactions (i.e., oxidation and oligomerization reactions) (Ervens et al., 2011; Gilardoni et al., 2016; Lee et al., 2013; Lei et al., 2024; Li et al., 2020; Powelson et al., 2014). In contrast, OOA formation primarily relies on gas-phase oxidation of VOCs with high-reactivity (i.e., aromatics and long-chain alkanes) (i.e., OH radicals), which has low concentrations in BBOA (Akagi et al., 2011; Jimenez et al., 2009; Shrivastava et al., 2017; Yokelson et al., 2007). This suggested that BBOA could be the important precursors for aqSOA instead of OOA via aqueous-phase reactions. These results were consistent with previous research and most of the observation data were within the triangle space (Bao et al., 2023; Kim et al., 2019; Paglione et al., 2020)." **(Lines 456-478).**

9. Lines 406-407. The authors emphasize the importance of aqueous-phase reactions, but how can the contribution of photochemical pathways be judged?

**Response:** In the revised manuscript, we have used $O_x$ observational data to analyze the contribution of photochemical pathways to SOA formation in this study, which reads "...In contrast, the odd oxygen ($O_x = O_3 + NO_2$) showed weak correlations with both OOA and aqSOA concentrations during the campaign ($p > 0.1$) (not shown). Although the average $O_x$ concentration was higher during PP (51.1 ± 19.6 ppb) than

CP ($36.9 \pm 14.0$ ppb), no significant correlations were observed in either period (not shown). These results suggested that photochemical reactions might played a limited role in SOA formation in this study." **(Lines 322-328).**

10. Lines 497-500. How was this result obtained and please provide further clarification on the mechanism?

**Response:** Thanks for the reviewer's comment and suggestion. In this study, the effect of boundary layer height was minimized by $Abs_{370,BrC,sec}/\Delta CO$ to characterize the evolution of secondary BrC absorption. During the campaign, the values of $Abs_{370,BrC,sec}/\Delta CO$ increased with the increases of aqSOA and ALWC concentrations from 17:00 to 03:00 LT ($r^2 = 0.63$, $0.57$, $p < 0.001$), when the average ALWC concentration (45 µg m$^{-3}$) was higher than that from 03:00 to 17:00 LT (39 µg m$^{-3}$) (Fig. 8). Previous research found that the SOA with strong absorption across UV to Vis range (i.e., guaiacol dimer and nitroguaiacol) could be formed via aqueous-phase reactions under high ALWC conditions (Gilardoni et al., 2016; Kroflic et al., 2015; Yang et al., 2021). This suggested secondary BrC chromophores with strong absorption at 370 nm were formed under the high ALWC from 17:00 to 03:00 LT. Additionally, we have provided further clarification on the mechanism of aqSOA with strong absorption formation from the aged BBOA via aqueous-phase reactions.

It has been revised accordingly in the revised manuscript, which reads "During the campaign, the relationship between $Abs_{370,BrC,sec}$ and SOA factors mass concentrations was analyzed to understand the correlation between secondary BrC absorption and its chromophores. As shown in Fig. S16, $Abs_{370,BrC,sec}$ significantly increased with the increase of aqSOA concentrations ($r^2 = 0.44$, $p < 0.001$) and high $Abs_{370,BrC,sec}$ values were consistent with the high ALWC values, this was not the case for OOA ($p > 0.1$).

The slope of the linear regression (3.50) between aqSOA mass concentrations and $Abs_{370,BrC,sec}$ was higher than OOA (Fig. S16), so was the MAC values of aqSOA across UV to Vis range (Fig. S14). To further characterize the evolution of secondary BrC absorption, $Abs_{370,BrC,sec}$ was normalized by $\Delta CO$ (the background-corrected CO mixing ratios) to minimize the effect of boundary layer height (Fig. 8) (DeCarlo et al., 2010). Here, the background CO value (400 ppb) was defined as the lowest 1.25th percentile of the CO values during the campaign (Kondo et al., 2006). Fig. 8 shows that the values of $Abs_{370,BrC,sec}/\Delta CO$ increased with the increases of aqSOA and ALWC concentrations from 17:00 to 03:00 LT ($r^2 = 0.63, 0.57, p < 0.001$), while $Abs_{370,BrC,pri}/\Delta CO$ slightly decreased with the increases of BBOA and m/z 60 concentrations ($r^2 = 0.35, 0.33, p < 0.001$). Additionally, the mass concentrations of $NO_3$, $NH_4$, and $NO_2$ from 17:00 to 03:00 LT were 1.2, 1.2, and 1.3 times that from 04:00 to 16:00 LT during the campaign. These results were similar to those observed in SCB during winter (Peng et al., 2025; Wu et al., 2024). As described in section 3.2, the SOA with hydroxyl groups (i.e., glyoxal and methylglyoxal) could be formed from the aged BBOA via aqueous-phase reactions under high ALWC during the campaign. Previous research have shown that oligomers (involving two glyoxal molecules) formed via aqueous reactions of glyoxal with $NH_3$ contain C=C or C=N bonds, exhibiting strong absorption at near-UV (Laskin et al., 2015; Lee et al., 2013; Nozière et al., 2009; Powelsonet al., 2014). This suggested secondary BrC chromophores with strong absorption at 370 nm were formed under the high ALWC from 17:00 to 03:00 LT, which might be related to the aqSOA from the aged BBOA via aqueous-phase reactions. The low values of $Abs_{370,BrC,sec}/\Delta CO$ at 12:00−14:00 LT could be related to the photolysis and/or photooxidation causing BrC photobleaching (Sareen et al., 2013; Zhao et al., 2015). Overall, we suggested that aqSOA formed

from biomass-burning emissions might be important for BrC absorption, especially at night."**(Lines 584-615).**

11. The conclusion section needs to be further condensed; it seems that parts of the conclusion have been emphasized several times, and there is a need for the author to reorganize the structure of the conclusion to give a complete and clear conclusion.

**Response:** It has been revised accordingly in the revised manuscript, which reads

**"4 Conclusions**

Field observations indicated that secondary organic aerosol (SOA) accounts for most of organic aerosol (OA) worldwide and aqueous-phase oxidation is an important pathway for the SOA formation. Our results demonstrated the fact that aqSOA was originated from the aged biomass-burning emissions via aqueous-phase reactions under high ALWC in the ambient atmosphere. Additionally, the less oxidized aqSOA formation via aqueous-phase reactions instead of photo-chemical reactions played a key role in the haze pollution dynamic evolution during the polluted period. This study also indicated that the impact on secondary BrC absorption should not be ignored, although primary BrC dominated the BrC absorption across ultraviolet to visible range. The aqSOA formed from aged biomass-burning emissions significantly contributed to the BrC budget and showed stronger absorption across ultraviolet to visible range than other OA components (except BBOA). The similarity between ambient data and the parameterized curve of $AAE_{370-880}$ versus BC-to-OA ratios in this study was consistent with the previous laboratory research on biomass-burning emissions. Higher values of $AAE_{370-880}$ and $MAC_{\lambda,aqSOA}$ reinforced that aqSOA formation from aged biomass-burning emissions via aqueous-phase reactions had stronger absorption than that via photo-chemically reactions.

In conclusion, our results revealed the aqSOA formation and brownness from aged BBOA via aqueous-phase reactions and highlighted the importance of aqSOA on aerosol pollution and absorption in the Sichuan Basin, China. Brown aqSOA originating from biomass-burning emissions was an important player in air quality budget and climate forcing balance worldwide. And it should be taken into account in air quality and climate models for a correct description of the global OA budget and its climate-relevant optical properties. This study was helpful in understanding the formation, light properties, and impacts of aqSOA in the ambient atmosphere. Future research should focus on the molecular-level characterization, transportation, and reactivities of gas and particle-phase aqSOA precursors to improve understanding of aqSOA formation processes and absorption properties." **(Lines 652-680).**

---

## Author Comment (AC2)

Dear Reviewers,

We appreciate the comments of reviewers and have revised the manuscript to address the review comments from all reviewers. We have provided a point-to-point response to the concerns, suggestions, and questions. We hope our work could satisfy both the reviewers and the editor. Our answers to each question have been marked in blue text. The annotated line numbers refer to the revised copy of the manuscript.

This manuscript by Peng et al. uses ambient measurements of organic aerosols (OA) chemical composition and aerosol light absorption to understand the relation between the sources of OA and their absorption properties. The authors claim, using positive matrix factorization on OA, that a significant fraction of secondary OA during pollution episodes in the Sichuan Basin originate from the aqueous processing of biomass burning OA in aerosol water. These secondary OA formed in the aqueous phase are further argued to contribute largely to the light absorption by secondary brown carbon.

While this study makes great use of different datasets and statistical methods to understand the relation between OA sources and aerosol light absorption during clean and polluted periods, the text contains a lot of repetitive information that are often not clearly expressed. I also do not understand why this manuscript has been submitted as a measurement report. Indeed, the scientific conclusions from this manuscript are not limited to a description of observations made during a measurement campaign, but are based on parameters calculated/inferred from these observations (e.g., using positive matrix factorization, or a multiple linear regression method).

Below is a list of major and minor comments on the manuscript that I believe should be considered by the authors to facilitate publication into ACP. A number of technical corrections, mostly regarding language, are also suggested for improved clarity.

**Major comments:**

1. The paper often uses terms such as "field observations", "direct observations", or "ambient observations" (e.g., lines 24, 266, 526, 551, 559, 575) to refer to aqSOA, its formation and its properties. This is misleading, as aqSOA was not "observed" but is the result of a statistical model (positive matrix factorization). The same goes for the light absorption by secondary brown carbon, which is a value derived from statistical methods (minimum R-squared method and multiple linear regression method). This needs to be clarified in the manuscript.

**Response:** Thanks for the reviewer's comment. We revised the manuscript to clarify this distinction and have updated the terminology to reflect that the values for aqSOA and light absorption by secondary brown carbon are derived from statistical models.

2. I find that a justification and a discussion of the PMF factors is missing in the manuscript. A lot of the discussion is linked to the PMF factors (mostly aqSOA and BBOA), but apart from a short paragraph in the SI where a couple of correlations to external tracers are given, there is little justification and description of the PMF factors. For instance, it is mentioned at lines 45-50 of the SI that the factors were constrained based on factors from the literature, but the description of the way the avalues were dealt with is confusing (values from 0.7 to 1 deleted for HOA and CCOA, but what about BBOA?).

**Response:** Thanks for the reviewer's comment. We have described the details of OA source apportionment procedures and present detailed diagnostic plots of the PMF results in the supporting information (SI Text S1 and Figures S4−S10). And the figures for the 4-6 factors solutions, the evolution the $Q/Q_{exp}$ for the different number of factors, and $a$-values were shown in Figures S4−S8.

"**Text S1.** Source Apportionment of OA

Here, the positive matrix factorization (PMF) and multilinear engine (ME2) were implemented on the OA data measured by ToF-ACSM to determine the numbers and types of OA source factors (Paatero 1999; Paatero and Tapper 1994). The data analyses were conducted using the source finder (SoFi, Canonaco et al., 2013) tool version 6.3 for Igor Pro. PMF is a bilinear unmixing model that can describe the variability of a multivariate database as a linear combination of static factor profiles and their corresponding time series, as expressed in Eq. (S1):

$$X=GF+E \tag{S1}$$

where X refers to the measured OA mass spectral matrix containing i rows and j columns, F contains the factor profiles, G their corresponding time series, and E is the model residuals and the scaled residuals are minimized. The model uses a least-squares method to iteratively minimize the object function Q, defined as the sum of the squared residuals ($e_{ij}$) weighted by their respective uncertainties ($\sigma_{ij}$):

$$Q=\sum_{i=1}^{m}\sum_{j=1}^{n}\left(\frac{e_{ij}}{\sigma_{ij}}\right)^2 \tag{S2}$$

The PMF analysis was performed using the ME2, which allows for a comprehensive exploration of rotational ambiguity in the solution space (Canonaco et al., 2013). ME2 is used to introduce a priori information as an additional model input, constraining one or more factor profiles to a predetermined range (*a* value):

$$f_{j,solution}=f_j \pm a \times f_j \qquad \text{(S3)}$$

where f refers to a row of the matrix F, j indicates the m/z of the ions, and the *a* value (from 0 to 1 with a step of 0.1) determines the extent to which the output profiles can differ from the model inputs.

The optimal number was selected by the discrimination of the tracers and the spectrum pattern of each source. Hydrocarbon-like OA (HOA) was dominated by alkyl ions with prominent ion fragments at $C_nH^+_{2n-1}$ and $C_nH^+_{2n+1}$ (m/z 41, 43, 55, 57, 69, 71, 83, and 85) in the spectra (Elser et al., 2016). Biomass-burning OA (BBOA) was identified by significant contributions from m/z 60 (mainly $C_2H_4O_2^+$) and m/z 73 (mainly $C_3H_5O_2^+$), they were the fragments of levoglucosan and mannosan emitted from incomplete biomass burning (Alfarra et al., 2007). Coal-combustion OA (CCOA) was characterized by unsaturated hydrocarbon ion fragments such as PAH-related ion fragments (i.e., m/z 77, 91, and 115) (Sun et al., 2016). Oxygenated OA (OOA) was distinguished by the prominent signal of m/z 43 (mainly $C_2H_3O^+$) and m/z 44 (mainly $CO_2^+$) (Ng et al., 2011). Aqueous-phase oxidized OA (aqSOA) also had high correlation with m/z 43 and m/z 44, while it might show a significantly higher m/z 29 (mainly $CHO^+$) signal than other OA factors (Zhao et al., 2019; Zhong et al., 2021).

In this study, we performed the free PMF runs from 3 to 6 factors. The discrete spectra were more likely to reasonable results when N = 4, 5, and 6 (Fig. S4−S6). When N = 4, OOA1, OOA2, HOA and BBOA+CCOA were resolved. In

BBOA+CCOA, the signals of levoglucosan (m/z 60) and PAH-related (m/z 91 and 115) ion fragments were obvious in the spectrum, suggesting BBOA was mixed with CCOA. In the N = 5 solutions, BBOA, CCOA, HOA, OOA, and aqSOA were resolved. These solutions were more reasonable than the N = 4 solutions because of the high relative humidity conditions during the sampling period. However, both CCOA and HOA exhibited significant signals at m/z 60, suggesting potential mixing with BBOA. When N = 6, the resolved OOA profiles were less evident than those in N = 5. Additionally, $Q/Q_{expected}$ showed that the solution with 5 factors was the most interpretable results (Fig. S7).

It should be noted that there was still a mix among POA factors in the five factors PMF solutions (Fig. S5). Initially, we tried to constrain COA by using COA profile in ME2 to identify if COA factor could be resolved from the OA sources. The COA was tried to constrain by using the profile from Elser et al. (2016). The restriction method ME2 was used to minimize PMF rotational ambiguity by the $a$-values from 0 to 1 with a step of 0.1 based on the reasonable result of PMF solutions. However, a large amount of blank values were shown, indicating COA was absent during the campaign. Then, all the combinations (BBOA and HOA, BBOA and CCOA, HOA and CCOA) were freely optimized by 11 $a$-values variables, respectively.

Here, BBOA was constrained by using the BBOA profile of Zhong et al. (2020), which was a clean BBOA profile from free PMF result. HOA and CCOA were constrained by using the HOA profile of Ng et al. (2011) and CCOA profile of Wang et al. (2017) to avoid mix with BBOA, respectively. Compared with previous research (Zhong et al., 2021), the optimal $a$-values solutions and ion fragments tracers were

used to optimize the results, achieving effective minimization of m/z 60 signals in both HOA and CCOA profiles. In order to minimize the mix of BBOA from HOA and CCOA, a threshold of m/z 60 (the major ion tracer of BBOA) for the maximal fractional contribution was set as 0.006 (mean + 2σ) for HOA and CCOA, respectively (Ng et al., 2011; Wang et al., 2017).

(1) When BBOA and HOA were freely combined in increments of 0.1 (from 0 to 1), the fractional contribution of m/z 60 in HOA ranged from 0.0017 to 0.0189. When the $a$-values of BBOA and HOA were set to ranges of 0−1 and 0−0.6, respectively, the fractional contribution of m/z 60 in HOA remained below 0.006. However, when the $a$-values of HOA extended to 0.7−1, the corresponding m/z 60 fractional contributions exceeded 0.006 (Fig. S8a). Therefore, only solutions with HOA $a$-values below 0.7 were retained, while all BBOA solutions with $a$-values ranging from 0 to 1 were retained.

(2) BBOA and CCOA were also freely combined by 11 $a$-values variables according to the $a$-values method in (1). When the $a$-values of CCOA were higher than 0.9, all fractional contributions of m/z 60 in CCOA were above 0.006 (Fig. S8b). In contrast, at CCOA $a$-values below 0.9, all the corresponding m/z 60 contributions in CCOA remained below this threshold (< 0.006). Therefore, the BBOA solutions with $a$-values ranging from 0 to 1 were retained, but all solutions with CCOA $a$-values exceeded 0.9 should be eliminated.

(3) After HOA and CCOA were constrained, the other unconstrained factors had a similar pattern with previously reported profiles. Based on the results from (1) and (2), we selected $a$-values ranges of 0−0.6 for HOA and 0−0.8 for CCOA for the factors combination. As shown in Fig. S8c, when the $a$-values of CCOA exceeded 0.7,

most of the corresponding m/z 60 fractional contributions were above 0.006. Therefore, both the HOA and CCOA solutions with *a*-values from 0.7 to 1 were deleted based on the aforementioned criteria. Overall, 20 PMF solutions were retained and their average was used as the final result (Fig. S9)."

3. For the absorption values, the authors often refer to measurements of $Abs_{BrC}$, but it is never clearly stated that this value is calculated from $Abs - Abs_{BC}$, where the latter is calculated based on assumptions stated at lines 183-184. A more thorough description of this is needed in the manuscript or in the SI, as it currently missing.

**Response:** Thanks for the reviewer's comment. We have clearly stated that the Abs was divided into BC and BrC absorption ($Abs_{BC}$ and $Abs_{BrC}$) and the Abs value was the sum of $Abs_{BC}$ and $Abs_{BrC}$ at different wavelengths. Then, $Abs_{BrC}$ value is calculated from $Abs - Abs_{BC}$. Additionally, a more thorough description of this is needed in the SI Text S2. It has been revised accordingly in the revised manuscript, which reads "The $Abs_\lambda$ was divided into BC and brown carbon (BrC, a group of colored OA compounds) absorption ($Abs_{\lambda,BC}$ and $Abs_{\lambda,BrC}$) ($Abs_\lambda=Abs_{\lambda,BC}+Abs_{\lambda,BrC}$) and characterized by the absorption Ångström exponents (AAE) (Laskin et al., 2015). Here, $Abs_\lambda$ was determined dependent $BC_\lambda$ mass concentrations ($Abs_\lambda=BC_\lambda\times MAC_\lambda$). We assumed the mass absorption cross-section of aerosols ($MAC_\lambda$) were 18.47, 14.54, 13.14, 11.58, 10.35, 7.77, and 7.19 $m^2$ $g^{-1}$ at 370, 470, 520, 590, 660, 880, and 950 nm, respectively (Drinovec et al., 2015; Zhu et al., 2017). Here, we assumed that $Abs_{880}$ was sole from BC, then the following formula was used to determine $Abs_{\lambda,BC}$ values: $Abs_{\lambda,BC}=Abs_{880}\times(880/\lambda)^{-AAE_{BC}}$ (Drinovec et al., 2015; Kirchstetter and Novakov, 2004; Moosmüller et al., 2009; Qin et al., 2018; Zhu et al., 2017). The AAE of BC ($AAE_{BC}$) value was obtained from the equality:

$AAE_{BC}=-\log(Abs_{880}/Abs_{950})\div\log(880/950)$ (Wang et al., 2021). A detailed description of $Abs_{\lambda,BC}$ and $Abs_{\lambda,BrC}$ calculations is provided in SI Text S3. Additionally, $Abs_{\lambda,BrC}$ was caused by primary and secondary BrC light absorption ($Abs_{\lambda,BrC,pri}$ and $Abs_{\lambda,BrC,sec}$). The $Abs_{\lambda,BrC,sec}$ value was calculated by a minimum R-squared (MRS) method at each wavelength (Wang et al., 2019; Wu and Yu, 2016; Wu et al., 2024). The detailed information of MRS method and $Abs_{\lambda,BrC,sec}$ estimation is provided in SI Text S3." **(Lines 214-231).**

"**Text S3.** Estimation of BC and BrC Absorption

Aerosol light absorption ($Abs_{\lambda}$) was caused by black carbon (BC) ($Abs_{\lambda,BC}$) and brown carbon (BrC) ($Abs_{\lambda,BrC}$). In this study, the $Abs_{\lambda}$ value was measured in real time by AE33 (Magee Scientific) at seven wavelengths (i.e., 370, 470, 520, 590, 660, 880, and 950 nm) (Drinovec et al., 2015). Previous research indicated that $Abs_{\lambda}$ measured by Aethalometer was influenced by the scattering effects of quartz filters and nonlinear loading effects (Coen et al., 2010). This study implemented real-time corrections for loading effects based on the parallel measurements of attenuation values (ATN1 and ATN2) from the AE33 (Drinovec et al., 2015).

Using equations (S5) and (S6), BC concentrations and the coefficient $k$ for eliminating filter scattering effects were calculated based on the measured ATN1 and ATN2 values, along with the mass concentrations of BC1 and BC2. The $Abs_{\lambda}$ was derived from BC concentrations at different wavelengths and the mass absorption cross-section of aerosols ($MAC_{\lambda}$) according to equation (S7) (Zhu et al., 2017). Then, $Abs_{\lambda,BC}$ and $Abs_{\lambda,BrC}$ were calculated based on the equations (S8), (S9), and (S10) (Qin et al., 2018; Wang et al., 2021; Zhu et al., 2017).

$$BC1=BC\times(1-k\times ATN1) \tag{S5}$$

$$BC2=BC\times(1-k\times ATN2) \tag{S6}$$

$$Abs_\lambda=BC_\lambda\times MAC_\lambda \tag{S7}$$

$$Abs_{BrC,\lambda 1}=Abs_{\lambda 1}-Abs_{BC,\lambda 1} \tag{S8}$$

$$Abs_{BC,\lambda 1}=Abs_{\lambda 2}\times(\lambda_2/\lambda_1)^{AAE_{BC}} \tag{S9}$$

$$AAE_{BC}=-\log(Abs_{880}/Abs_{950})\div\log(880/950) \tag{S10}$$

where $Abs_{\lambda 2}$ represented the Abs at 880 nm. We assumed that $Abs_{BrC}$ was considered negligible at 880 nm, and $MAC_\lambda$ values were 18.47, 14.54, 13.14, 11.58, 10.35, 7.77, and 7.19 $m^2\ g^{-1}$ at 370, 470, 520, 590, 660, 880, and 950 nm, respectively (Drinovec et al., 2015; Qin et al., 2018; Zhu et al., 2017).

$Abs_{\lambda,BrC}$ was caused by primary BrC ($Abs_{\lambda,BrC,pri}$) and secondary BrC ($Abs_{\lambda,BrC,sec}$). The $Abs_{\lambda,BrC,sec}$ value was estimated by a minimum R-squared (MRS) method at each wavelength developed from the BC-tracer method (Wang et al., 2019; Wu and Yu, 2016) ..."

4. The manuscript contains a lot of redundant sentences and grammatically incorrect sentences that perturb the reading. I have tried listing most of these issues below in the "minor comments and technical corrections" section.

**Response:** Thanks for the reviewer's comment and suggestion. We have provided a point-to-point response to the following minor comments and technical corrections. Additionally, all sentences have been grammatically verified and improved in the revised manuscript.

**Minor comments and technical corrections:**

Lines 40-42: I would suggest changing to present tenses when you make a general statement not necessarily related to one particular study. This comment is valid elsewhere in the manuscript (e.g., lines 49, 76, 85, etc.). Please check.

**Response:** It has been revised accordingly in the revised manuscript.

Lines 52-56: This sentence is very difficult to understand as it is. Can you maybe split it in two and reformulate it?

**Response:** Thanks for the reviewer's suggestion. It has been revised accordingly in the revised manuscript, which reads "An increasing number of studies pointed toward aqSOA as a major SOA could form in fogs, clouds, and aerosol water (Ervens et al., 2011; Ortiz-Montalvo et al., 2012; Tan et al., 2012; Xu et al., 2022). And the oxygenated VOCs (OVOCs) with large water-soluble and low Henry's constant (i.e., methylglyoxal and glycolaldehyde) are the important aqSOA precursors (Ortiz-Montalvo et al., 2012; Tan et al., 2012)." **(Lines 52-56).**

Line 94: What do you mean by "typical city in SCB"? This comment is also valid for the statement at line 106. For the general reader, it is not clear what "typical" means in that sense. Is it typical in terms of aerosol pollution, in terms of population, in terms of meteorological conditions?

**Response:** Thanks for the reviewer's comment. It has been revised accordingly in the revised manuscript, which reads "Here a time-of-flight aerosol chemical speciation monitor (ToF-ACSM) and a series of collocated instruments were used to characterize aqSOA dynamic evolution from biomass burning under real ambient conditions in a typical city with relatively serious air pollution in SCB from October 21 to November

23, 2022." **(Lines 105-108)**, "An intensive field campaign on the chemical and physical properties of aerosol was conducted at a measurement site in a city affected by severe aerosol pollution (Yongchuan, 29°21′25″ N, 105°54′6″ E) from October 21 to November 23, 2022. This is a typical urban site surrounded by restaurants, shopping malls, and residential buildings, and the site is located in a parallel ridge-and-valley area between two megacities in SCB (Chongqing center and Chengdu) (Fig. S1). It was primarily influenced by multiple local emissions from traffic (arterial roads to the east 600 m and west 300 m) and a variety of residential sources (i.e., biomass burning and fossil fuel combustion)." **(Lines 117-125).**

Lines 104-105: Please rephrase (e.g., "… was conducted at a measurement site in a city affected by severe aerosol pollution").

**Response:** It has been revised accordingly in the revised manuscript, which reads "An intensive field campaign on the chemical and physical properties of aerosol was conducted at a measurement site in a city affected by severe aerosol pollution (Yongchuan, 29°21′25″ N, 105°54′6″ E) from October 21 to November 23, 2022." **(Lines 117-119).**

Line 111: What do you mean by "interfered by neighboring buildings"? Do you mean in terms of dynamics or emission sources?

**Response:** It has been revised accordingly in the revised manuscript, which reads "There was no interference of dynamics from neighboring buildings, and measurements at the site helped understand the characteristics of haze pollution dynamic evolution." **(Lines 125-127).**

Lines 120-125: I would suggest moving this part to Sect. 2.3.1 where it discussed in more details at lines 146-150.

**Response:** Thanks for the reviewer's suggestion. We have moved this part to Sect. 2.3.1, which reads "The ionization efficiency (IE) and relative ionization efficiency (RIEs) were regularly calibrated using a scanning mobility particle sizer with a differential mobility analyzer (SMPS 3081A, TSI) and a condensation particle counter (CPC 3775, TSI). The comprehensive overview of the operation and calibration procedures of ToF-ACSM can be found in Bao et al. (2023)." **(Lines 156-161).**

Line 121: Please change to "… were regularly calibrated using a scanning…"

**Response:** It has been revised accordingly in the revised manuscript.

Line 124: Please change "could" to "can".

**Response:** It has been revised accordingly in the revised manuscript.

Line 127: The Aethalometer AE33 measures equivalent black carbon, not black carbon. Please clarify that in the text.

**Response:** It has been revised accordingly in the revised manuscript, which reads "A seven-wavelength Aethalometer (AE33, Magee Scientific) was used to measure the aerosol light absorption (Abs$_\lambda$) and equivalent black carbon (BC$_\lambda$) mass concentrations in real time at 370, 470, 520, 590, 660, 880, and 950 nm." **(Lines 136-138).**

Line 135: Please change "could" to "can".

**Response:** It has been revised accordingly in the revised manuscript.

Line 157: Please spell out the "SoFi" acronym (i.e., Source Finder).

**Response:** It has been revised accordingly in the revised manuscript.

Line 163: I would suggest adding the reference of Paatero and Hopke (2003), as they originally presented this downweighting scheme.

**Response:** Thanks for the reviewer's suggestion. It has been revised accordingly in the revised manuscript, which reads "The ions data with signal-to-noise (S/N) lower than 0.2 were discarded, and those S/N from 0.2−2 were downweighted by a factor of 2 (Bao et al., 2023; Paatero and Hopke 2003)." **(Lines 189-191).**

Lines 183-184: The formulation "Abs880 was sole from BC" is not clear. Do you mean that it is assumed that, at 880 nm, only BC is contributing? Please rephrase.

**Response:** It has been revised accordingly in the revised manuscript, which reads "Here, we assumed that $Abs_{880}$ was sole from BC, then the following formula was used to determine $Abs_{\lambda,BC}$ values: $Abs_{\lambda,BC}=Abs_{880}\times(880/\lambda)^{-AAE_{BC}}$ (Drinovec et al., 2015; Kirchstetter and Novakov, 2004; Moosmüller et al., 2009; Qin et al., 2018; Zhu et al., 2017)." **(Lines 220-223).**

Lines 195-198: If I understand correctly, the constant a-e are wavelength-dependent right? If so, can you clarify that?

**Response:** Thanks for the reviewer's comment. Yes, the constant a-e are wavelength-dependent. It has been revised accordingly in the revised manuscript and the Abs of five OA factors at each wavelength obtained by the MLR method were shown in Table S1, which reads "… the a–e were constants, used to optimize the $Abs_\lambda$ of each OA component, and equivalent to MAC values at each wavelength (i.e., a–e at 370

nm represented $MAC_{370,OOA}$, $MAC_{370,BBOA}$, $MAC_{370,CCOA}$, $MAC_{370,aqSOA}$, and $MAC_{370,HOA}$, respectively). " **(Lines 236-239).**

Line 209: Similar to a comment I made above, please use present or present perfect tense when referring to a figure shown in the main or SI (i.e., "… during the campaign is shown in Fig. 1."). This comment is valid elsewhere in the manuscript. Please check.

**Response:** It has been revised accordingly in the revised manuscript.

Lines 216-217: There is an alternate use of "Org" and "OA" in this manuscript. While I understand that "Org" refers to the species measured by the ACSM, and "OA" is more general, the distinction is not always clear. I would suggest sticking to one or another.

**Response:** It has been revised accordingly in the revised manuscript.

Lines 220-222: As it is written, giving a range of mass concentrations does not justify that the "OA composition were substantially different". Either refer to Fig. 2 in this sentence, or move this sentence to the next paragraph and merge it with the sentence where you actually discuss the differences in chemical composition.

**Response:** Thanks for the reviewer's comment. We have moved and revised this sentence to the next paragraph, which reads "During PP, the mass concentrations of BC+NR-$PM_{2.5}$ and OA were $102.3 \pm 26.9$ and $57.4 \pm 22.5$ $\mu g\ m^{-3}$, 2.5 and 3.1 times that during CP, respectively. As shown in Fig. 2, the $PM_{2.5}$ species were substantially different in the PP and CP." **(Lines 270-273).** Additionally, the substantially different of OA composition was pointed out in the revised manuscript, which reads "It should be noted that the contributions of BBOA and aqSOA to OA increased from CP

(31.7% and 12.6%) to PP (38.6% and 14.1%), while CCOA, HOA, and OOA contributions decreased." **(Lines 298-300).**

Line 227: Since you refer to the method as "PMF" before, stick to "PMF" and not "PMF-ME2".

**Response:** It has been revised accordingly in the revised manuscript.

Line 229: How can you tell for sure it is $C_2H_4O_2^+$ contributing at m/z 60, when you only have unit mass resolution with the ACSM? There are other ion fragments at this m/z. Please change to m/z 60, since you also just mention m/z 73 just after. You can add the ion fragment that is most likely contributing in parenthesis. This is valid elsewhere in the manuscript (e.g., line 251, 447, 448).

**Response:** It has been revised accordingly in the revised manuscript.

Line 229: This is the first time you mention a "correlation". Please define the metric you use. You mention "$r^2$", is it a Pearson correlation squared, a Spearman correlation squared, a coefficient of determination?

**Response:** Thanks for the reviewer's suggestion. Here, $r^2$ is a Pearson correlation squared. It has been revised accordingly in the revised manuscript.

Line 228-234: For someone who is not an absolute expert in PMF, this whole paragraph gives very little information on the justification of the choice of factor. At least give references for why the correlation mentioned justify the factor names.

**Response:** Thanks for the reviewer's comment. It has been revised accordingly in the revised manuscript, which reads "Here, five OA factors were identified by the PMF model with detailed information in SI Text S1, and the mass spectrum of these factors

is shown in Fig. S9. HOA mass spectrum was characterized by alkyl fragment ion series at $C_nH^+_{2n-1}$ and $C_nH^+_{2n+1}$ (i.e., m/z 41, 43, 55, and 57), they are common characteristics of primary combustion emissions (Elser et al., 2016; Lanz et al., 2007). BBOA was identified by the high signal of m/z 60 (mainly $C_2H_4O_2^+$) and m/z 73 (mainly $C_3H_5O_2^+$), they are the fragments of levoglucosan and mannosan emitted from incomplete biomass burning (Alfarra et al., 2007). CCOA had high correlation with the unsaturated hydrocarbon ion fragments such as PAH-related ion fragments (i.e., m/z 77, 91, 115), emitted from traditional coal combustion (Sun et al., 2016a). OOA was distinguished by the prominent signal of m/z 44 (mainly $CO_2^+$) and highly correlated with the oxygenated ions (Ng et al., 2011b). aqSOA also had high correlation with the the oxygenated ions (i.e, m/z 43 (mainly $C_2H_3O^+$) and m/z 44). Moreover, The mass spectrum of aqSOA showed a significantly higher m/z 29 (mainly $CHO^+$) signal than other OA factors, consistent with those reported in the previous studies (Sun et al., 2016a; Xu et al., 2019; Zhao et al., 2019; Zhong et al., 2021). Moreover, BBOA showed significant correlations with m/z 60 (mainly $C_2H_4O_2^+$) and m/z 73 (Pearson's $r^2$ ($r^2$) = 0.85, 0.80, $p < 0.001$); CCOA was strongly correlated with Chl and m/z 115 ($r^2$ = 0.56, 0.48, $p < 0.001$); HOA was correlated with $NO_2$ and m/z 41 ($r^2$ = 0.47, 0.59, $p < 0.001$); OOA and aqSOA were significantly correlated with $NO_3$, $NH_4$ ($r^2$ = 0.77, 0.75, $p < 0.001$) and $SO_4$, ALWC ($r^2$ = 0.67, 0.85, $p < 0.001$), respectively (Fig. S10). These results highlighted the result of five OA factors was reasonable." **(Lines 275-297).**

Figure 2: The legends are very confusing, as the elements of the legend in (a) also apply for (b) and vice-versa. The left y-axis are generally weird, can you make them start from zero and have constant increments?

**Response:** Thanks for the reviewer's suggestion. It has been revised accordingly in the revised manuscript.

[Figure]

Lines 242-245: Please reformulate this sentence, it is grammatically incorrect.

**Response:** It has been revised accordingly in the revised manuscript, which reads "Compared with CP, the obvious diurnal variation of OA concentration was exhibited during PP. As shown in Fig. 2, the OA concentration peak (82.7 µg m$^{-3}$) was observed at 12:00 local time (LT) in the daytime during PP, while observed at 21:00 LT at night during CP." **(Lines 306-309).**

Line 257-258: Do you rather mean here the chemical processing from BBOA to aqSOA? Since they don't peak at the same time the argument was more that one leads to the other, not that there are processed at the same time.

**Response:** It has been revised accordingly in the revised manuscript, which reads "…supporting that ALWC might play a significant role in the chemical processing of aqSOA formation from BBOA during PP." **(Lines 321-322).**

Line 261-265: This sentence is a repetition of the sentence at the end of the last paragraph. Also, you now mention fog events, but you mentioned before that the aqueous processing was related to ALWC, so aerosol water, not fog water. Regarding the last point, you mention this difference at lines 305-308, so you could move it here.

**Response:** Thanks for the reviewer's suggestion. It has been revised accordingly in the revised manuscript, which reads "Additionally, considerable aqSOA in the daytime during PP might be related to the high aerosol water and BBOA emissions in the harvest season − autumn − in SCB (Bao et al., 2023; Chen et al., 2017; Chen et al., 2019; Tao et al., 2014). Based on the direct observation of aqSOA, Gilardoni et al. (2016) also found that aqSOA such as guaiacol dimer ($C_{14}H_{14}O_4^+$) could be formed from aged biomass-burning emissions at both in fog water and in wet aerosol, especially under high ALWC conditions." **(Lines 331-337).**

Lines 281-283: Please reformulate this sentence, it is grammatically incorrect.

**Response:** Thanks for the reviewer's suggestion. It has been revised accordingly in the revised manuscript, which reads "Fig. 3 shows the strong correlation between the mass fraction (%) of aqSOA in total $PM_{2.5}$ and ALWC during the campaign ($r^2 = 0.64$, $p < 0.001$). The contribution of aqSOA increased with the increase of $f_{29}$ values (normalized mass spectrum signal at m/z 29)." **(Lines 352-355).**

Figure 3: Is the f29 coming from the ambient ACSM data? This distinction was never made and it is a bit confusing, as you sometimes also refer to the f29 for each induvial

factor separately (which are constant values and not variables). Also, why is the x-axis the percentage of aqSOA to BC+NR-PM$_{2.5}$ and not the percentage of aqSOA to total PMF OA as shown in Fig. 1d.

**Response:** In Fig. 3, the $f_{29}$ (normalized mass spectrum signal at m/z 29) came from the mass spectral matrix of OA for m/z 10−120, which are variables, and the time resolution of $f_{29}$ was 1 h for the continuous measurements. As the reviewer pointed out, the $f_{29}$ values for each individual factor (which are constant values) are sometimes used separately in the manuscript. These values are derived from the mass spectra of six OA factors during the campaign (Fig. S9). In order to analyze the direct impacts of ALWC and aqSOA on PM$_{2.5}$ formation, we conducted a correlation analysis using the percentage of aqSOA to BC+NR-PM$_{2.5}$ as the x-axis and ALWC as the y-axis.

Line 288: What do you mean by "located in a similar region"? Which figure are you referring to here?

**Response:** Thanks for the reviewer's comment. It has been revised accordingly in the revised manuscript, which reads "Moreover, both aqSOA concentrations and $f_{29}$ were well correlated with ALWC ($r^2 = 0.85, 0.73, p < 0.001$) (Fig. 3). During the campaign, the average value of the oxygen-to-carbon ratio (O:C) of aqSOA factor (0.85) was 2.7 times that (0.31) of BBOA factor. However, the similar hydrogen-to-carbon ratio (H:C) values of aqSOA factor and BBOA factor were observed (1.74 and 1.81, respectively), indicating that a hydrogen atom might be replaced by a OH moiety (Lim et al., 2010; Ng et al., 2011a). These results were similar to aqSOA observed in Italy and Beijing (Gilardoni et al., 2016; Zhao et al., 2019)." **(Lines 35-365).**

Lines 292-296: The correlations mentioned are not shown on any figure, please add "(not shown)" in the sentence.

**Response:** It has been revised accordingly in the revised manuscript.

Line 295: Change "production" to "produced".

**Response:** It has been revised accordingly in the revised manuscript.

Lines 301-305: Please reformulate this sentence, it is grammatically incorrect.

**Response:** It has been revised accordingly in the revised manuscript, which reads "It should be noted that only aqSOA concentrations were enhancement under high ALWC conditions (> 100 µg m$^{-3}$). It is likely because more water-soluble organic species (i.e., glyoxal and methylglyoxal) were formed, that were further oxidized to form aqSOA via aqueous-phase reactions in aerosol liquid water (Carlton et al., 2007; Ervens et al., 2011; Tan et al., 2012)." **(Lines 380-384).**

Line 311: This is the first time you make a distinction between POA and SOA. What exactly do you consider as POA and SOA for you PMF factors? You need to make this clear here as the reader doesn't know what you are referring to otherwise. At line 361, you explain what you mean by "SOA" but still not for POA.

**Response:** It has been revised accordingly in the revised manuscript, which reads "…although POA (BBOA+CCOA+HOA) and SOA (OOA+aqSOA) contributions were fairly constant across different ALWC levels (58−68% and 32−42%)." **(Line 387-389).** Moreover, the POA and SOA for the PMF factors in this study have been defined in the revised manuscript, which reads "Finally, five OA factors with function of the rotational parameter ($f_{peak}$ = 0) were identified, including three POA factors (i.e., BBOA, coal-combustion OA (CCOA), and hydrocarbon-like OA (HOA)) and two SOA factors (i.e., OOA and aqSOA) (Fig. S9 and S10)." **(Lines 191-195).**

Lines 341-342: This sentence feels like a repetition of what was said just before with Fig. 4.

**Response:** Thanks for the reviewer's suggestion. It has been revised accordingly in the revised manuscript, which reads "The correlation $r^2$ between aqSOA and BBOA concentrations was higher during PP (0.64) than that during CP (0.54) ($p < 0.001$) (Fig. 5a and c). Though aqSOA and BBOA concentrations increased with the increase of ALWC during CP and PP, the correlations between ALWC and aqSOA or BBOA concentrations were relatively stronger during PP than that during CP ($p < 0.001$)." **(Lines 418-422).**

Lines 342-345: The comparison is difficult to make, as you have much less datapoints during polluted periods. Can you check the statistical significance of your statement of "stronger positive correlation"?

**Response:** It has been revised accordingly in the revised manuscript, which reads "The correlation $r^2$ between aqSOA and BBOA concentrations was higher during PP (0.64) than that during CP (0.54) ($p < 0.001$) (Fig. 5a and c). Though aqSOA and BBOA concentrations increased with the increase of ALWC during CP and PP, the correlations between ALWC and aqSOA or BBOA concentrations were relatively stronger during PP than that during CP ($p < 0.001$)." **(Lines 418-422).**

Line 345: The correlations mentioned here are different than the ones given at lines 254-255 for the same variables. Why are they different? If these are supposed to be the same correlations, why repeat it here?

**Response:** These correlations are supposed to be the same correlations. It has been revised accordingly in the revised manuscript.

The correlations mentioned in **lines 317-319** "Additionally, the correlation between ALWC and aqSOA concentrations ($r^2 = 0.86$, $p < 0.001$) was stronger than BBOA concentrations ($r^2 = 0.58$, $p < 0.001$)…" indicated the ALWC played a more significant role in the aqSOA formation than BBOA during PP. In order to identify the coupling relationship between aqSOA and BBOA concentrations under different $PM_{2.5}$ pollution levels, We repeated it and compared the differences between CP and PP in **lines 419-422** "Though aqSOA and BBOA concentrations increased with the increase of ALWC during CP and PP, the correlations between ALWC and aqSOA or BBOA concentrations were relatively stronger during PP than that during CP ($p < 0.001$)."

Lines 345-347: Such statements that BBOA leads to aqSOA formation are repeated in almost every paragraph and it feels redundant (e.g., lines 290-291, 326-328, 350-352, 406-407), can you remove it in some places to avoid redundancy?

**Response:** It has been revised accordingly in the revised manuscript.

Lines 347-348: Figure 5b and d don't show any correlation between f29 and aqSOA, at least not using circle size for f29.

**Response:** Thanks for the reviewer's suggestion. We have revised accordingly in the revised manuscript and showed the correlations between $f_{29}$ and aqSOA in Figure 5b and d.

[Figure]

Lines 348-350: If you mention that the correlation is "strong" can you give numbers? It also seems to be based on a subset of a subset of data, which doesn't seem so fair.

**Response:** Thanks for the reviewer's comment. The correlation between aqSOA and OOA concentrations during PP has been shown in Figure 5d. We have explained the reason for selecting data under the conditions of ALWC > 80 µg m$^{-3}$ and $f_{29}$ > 0.15 to analyze the correlation between OOA and aqSOA concentrations in this study. It has been revised accordingly in the revised manuscript, which reads "A few data points with high aqSOA and OOA concentrations had low $f_{29}$ values (0.071−0.102) in Fig. 5d, while the average value of $f_{44}$ (normalized mass spectrum signal at m/z 44) of these data points (0.103 ± 0.024) was 1.3 times that of all data points (0.080 ± 0.035) during PP. It was likely due to the formation of more-oxidized OOA under high ALWC values (> 80 µg m$^{-3}$) in this study (Xu et al., 2017). Previous research found

that the $f_{29}$ values of polyols and species with non-acid OH groups from biomass-burning emissions were lower than 0.15 (Canagaratna et al., 2015; Gilardoni et al., 2016; Zhao et al., 2014). Moreover, the mass fraction of aqSOA showed a stable increasing trend and remained high levels (from 18% to 22%) at ALWC > 80 μg m$^{-3}$, which was associated with a corresponding decrease in OOA (from 15% to 10%) (Fig. 4b). Compared with OOA ($p > 0.1$), the aqSOA concentrations showed strong positive correlation with ALWC ($r^2 = 0.73$, $p < 0.001$) when ALWC > 80 μg m$^{-3}$ during PP. In contrast, ALWC showed the weak correlations with aqSOA and OOA concentrations during CP ($p > 0.1$). It should be noted that a strong anticorrelation between aqSOA and OOA concentrations was observed during PP at ALWC > 80 μg m$^{-3}$ when $f_{29} > 0.15$ ($r^2 = 0.76$, $p < 0.001$), but not during CP ($p > 0.1$) (Fig. 5b and d). These results indicated that the aqSOA formation was more intensive than OOA at high ALWC levels during PP." **(Lines 423-441).**

Lines 355-359: Please reformulate this sentence, it is grammatically incorrect.

**Response:** Thanks for the reviewer's suggestion. It has been revised accordingly in the revised manuscript, which reads "Additionally, m/z's 44 and 43 are usually from different functional groups and the ratio changes as a function of atmospheric aging. The triangle plot of $f_{44}$ versus $f_{43}$ has been widely used to characterize OA evolution, and $f_{44}$ versus $f_{60}$ is commonly used to investigate the aging trend of BBOA (Ortega et al., 2013; Paglione et al., 2020; Xu et al., 2017; Xu et al., 2019)." **(Lines 445-450).**

Figure 6: The colors and the legend of the figure are very confusing. Especially because the color bar for ALWC has the same colors as for the PMF factors. It is also not clear what are the data points from other studies and what are the data points from this study since the legend only describe the squares but not the circles.

**Response:** Thanks for the reviewer's comment. It has been revised accordingly in the revised manuscript, which reads "**Figure 6.** Triangle plots of **(a, c)** $f_{44}$ (normalized mass spectrum signal at m/z 44) versus $f_{43}$ (normalized mass spectrum signal at m/z 43), and **(b, d)** $f_{44}$ versus $f_{60}$ (normalized mass spectrum signal at m/z 60) colored by ALWC (circles) during clean period and polluted period. The dashed lines in **(a)** and **(c)** were derived from Ng et al. (2010) and used to follow the aging of OA components in the atmosphere. The background space ($f_{60} < 0.003$) without biomass burning influence was also shown by the grey shaded area. The background value of secondary aged OA (brown dashed line) and the black dashed lines characterising the aging of BBOA in **(b)** and **(d)** were derived from Cubison et al. (2011). The data points (squares) included the measurements in this study (bordered in red) and previous research (Bao et al., 2023; Gilardoni et al., 2016; Kim et al., 2019; Ng et al., 2011a; Paglione et al., 2020; Xu et al., 2015; Xu et al., 2017; Xu et al., 2019; Zhao et al., 2017; Zhao et al., 2019). $f_{43}$ (mainly $C_2H_3O^+$) is a tracer for POA and fresh SOA. $f_{44}$ is a proxy of the OA oxygenation degree and used as a tracer for aged SOA. $f_{60}$ is a proxy of anhydrosugars emitted from biomass burning." **(Lines 507-519).**

Lines 361-363: The conclusion of this sentence is too far-fetched. How does a larger f44 for SOA compared to POA justifies the statement that aging of OOA leads to formation of aqSOA?

**Response:** Thanks for the reviewer's comment. It has been revised accordingly in the revised manuscript, which reads "However, the $f_{44}$ of SOA factors (i.e., OOA and aqSOA) (0.118 and 0.117) were observably higher than POA factors, showing the freshly oxidized properties of SOA. Meanwhile, $f_{44}$ of aqSOA was close to that observed in fogs (Gilardoni et al., 2016; Kim et al., 2019), highlighting the presence

of aqueous-phase reactions in this study. The relative abundance of m/z 45 (mainly $HCO_2^+$), a tracer ion for carboxylic acids, was higher in the aqSOA spectra than in the OOA spectrum (Fig. S9). It was consist with previous research which found that aqueous-phase reactions were important sources of oxygenated organic compounds, including organic acids (Ervens et al., 2011; Kim et al., 2019; McNeill, 2015; Sun et al., 2010; Yu et al., 2014)." **(Lines 452-461).**

Line 365: Do you mean Fig. 6b or maybe 6d, instead of Fig. 4c?

**Response:** It has been revised accordingly in the revised manuscript, which reads "Fig. 6b shows BBOA and aqSOA with higher $f_{60}$ values (0.019 and 0.011) than CCOA (0.009) and HOA (0.008)." **(Lines 461-462).**

Lines 365-367: Is 0.011 "much higher" than 0.009 or 0.008?

**Response:** It has been revised accordingly in the revised manuscript, which reads "Fig. 6b shows BBOA and aqSOA with higher $f_{60}$ values (0.019 and 0.011) than CCOA (0.009) and HOA (0.008)." **(Lines 461-462).**

Line 367: Insert comma after 0.002.

**Response:** It has been revised accordingly in the revised manuscript.

Lines 369-371: The information in parenthesis "f44 (representation of aged OA)" and "f60 (presence of anhydrosugars)" is important to understand Fig. 6. This information should come much before when introducing Fig. 6 (i.e., between line 353 and 359).

**Response:** Thanks for the reviewer's suggestion. It has been revised accordingly in the revised manuscript, which reads "Previous research demonstrated that $f_{44}$ (representation of aged OA) could be used as a tracer of aged SOA, $f_{43}$ (normalized

mass spectrum signal at m/z 43) as a tracer of POA and fresh SOA, and $f_{60}$ (presence of anhydrosugars) as a tracer of BBOA (Cubison et al., 2011; Ng et al., 2010)." **(Lines 442-445).**

Lines 373-376: You mention "previous research" but don't cite papers. Also, the second part of the sentence after the comma is unrelated to the first part. I.e., the fact that the observations are within the triangle space does not imply that POA are fresh and aqSOA are more oxidized and formed from BBOA.

**Response:** Thanks for the reviewer's comment. It has been revised accordingly in the revised manuscript, which reads "These results were consistent with previous research and most of the observation data were within the triangle space (Bao et al., 2023; Kim et al., 2019; Paglione et al., 2020)." **(Lines 476-478).**

Lines 379-382: What are the numbers in parenthesis after CP? Please reformulate the sentence, at present it is not possible to understand it.

**Response:** It has been revised accordingly in the revised manuscript, which reads "Compared with CP ($r^2 = 0.17$, slope $= -0.53$), $f_{44}$ showed a more significant increase as the decreasing of $f_{43}$ with higher $r^2$ value (0.70) and the regression slope of $f_{44}$ versus $f_{43}$ ($-1.09$) was closer to $-1$ during PP." **(Lines 481-484).**

Lines 383-387: I would argue that not "some" points are outside the bottom boundary of the triangle, but most are. Also, please rephrase the sentence, it is not grammatically correct.

**Response:** Thanks for the reviewer's suggestion. It has been revised accordingly in the revised manuscript, which reads "It should be noted that the points of $f_{44}$ versus $f_{43}$ were inside the upper boundary of the triangle region, and most points were outside

the bottom boundary of the triangle region during PP. These results suggested that less oxidized SOA were formed via aqueous-phase reactions instead of photo-chemical reactions during PP (Kim et al., 2019; Zhao et al., 2019)." **(Lines 485-489).**

Lines 387-389: In general, it also just seems that there is much less ALWC during CP (which makes sense since aerosol concentrations are lower during CP). I am not convinced that this is an argument to justify the sentence that follows.

**Response:** The following sentence has been removed in the revised manuscript.

Lines 389-391: Please reformulate this sentence, it is grammatically incorrect.

**Response:** This sentence has been removed to avoid redundancy in the revised manuscript.

Lines 394-398: This sentence is already in the caption of Fig. 6. No need to duplicate the information, remove it here.

**Response:** It has been revised accordingly in the revised manuscript.

Lines 398-399: This sentence should go in the caption and be remove here.

**Response:** It has been revised accordingly in the revised manuscript.

Lines 399-401: It is not clear if you are referring to the PMF factors or the datapoints. If you are referring to the PMF factors then this sentence is in contradiction with the sentence at lines 367-368, where it is said that OOA has an f60 below the 0.003 threshold.

**Response:** Thanks for the reviewer's comment. It has been revised accordingly in the revised manuscript, which reads "Except for several points, the $f_{60}$ values were

ubiquitously higher than 0.003, and most points fell in the triangular region, suggesting the contribution of biomass burning to OA." **(Lines 494-496).**

Lines 401-403: The part "while the mean f44 value was significantly higher" was already stated at lines 377-378.

**Response:** It has been revised accordingly in the revised manuscript, which reads "During PP, the $f_{60}$ values ranging from 0.005 to 0.019 (0.010 ± 0.004) were similar with CP (from 0.004 to 0.019, 0.010 ± 0.003)." **(Lines 496-498).**

Lines 403-405: I am bit skeptical about this sentence since the schematic space is indeed a schematic, so you cannot quantify how many of your datapoints are within this region. Perhaps formulating it differently would help.

**Response:** Thanks for the reviewer's suggestion. It has been revised accordingly in the revised manuscript, which reads "The correlation $r^2$ between $f_{44}$ and $f_{60}$ was higher during PP (0.72) than that during CP (0.31) ($p < 0.001$). Moreover, compared with all data points during PP, those in the schematic space of aged BBOA showed relatively higher ALWC, a pattern that differed from observations during CP." **(Lines 498-501).**

Lines 425-427: Are you making a general statement or an observation based on your data? Also, please reformulate the sentence, it is grammatically incorrect.

**Response:** It has been revised accordingly in the revised manuscript, which reads "Previous research indicated that OA from fresh and aged biomass-burning emissions, which exhibited absorption properties across UV to Vis range with significantly higher AAE value than BC, might contribute to a net positive radiative forcing (Laskin et al., 2015)." **(Lines 521-524).**

Line 431: Please remove "As shown in Fig. S6". The average value of $Abs_{370,BrC}$ is not shown in Fig. S6 and cannot be guessed since BrC is split between primary and secondary contribution.

**Response:** It has been revised accordingly in the revised manuscript.

Lines 434-436: The distinction between POA and primary brown carbon is unclear here. These are not the same thing and should not be interchanged.

**Response:** Thanks for the reviewer's suggestion. It has been revised accordingly in the revised manuscript, which reads "The $Abs_{\lambda,BrC,pri}$ and $Abs_{\lambda,BrC,sec}$ accounted for 56.8%−72.5% and 27.5%−43.2% of $Abs_{\lambda,BrC}$ from 370 nm to 660 nm respectively, indicating primary emissions were the main contributors to BrC absorption (Fig. S12)." **(Lines 530-533).**

Line 438: Change "Here we showed" to "Hereafter, we show".

**Response:** It has been revised accordingly in the revised manuscript.

Lines 443-446: Where do the values in parenthesis come from?

**Response:** It has been revised accordingly in the revised manuscript, which reads "As described in section 2.3.3, the Abs of five OA factors at each wavelength were obtained by the MLR method (Table S1). Compared with CCOA ($Abs_{370,CCOA}$), HOA ($Abs_{370,HOA}$), and OOA ($Abs_{370,OOA}$) (11.5%, 9.1%, and 11.1%), the Abs at 370 nm calculated for BBOA ($Abs_{370,BBOA}$) and aqSOA ($Abs_{370,aqSOA}$) showed higher contributions (51.9% and 16.4%) to $Abs_{370,BrC}$, consistent with the higher MAC values (Fig. S14)." **(Lines 539-544).**

**Table S1.** The Abs values and contributions of five OA factors to $Abs_{BrC}$ at different

wavelengths during the campaign.

| OA factors | Parameter | 370 nm | 470 nm | 520 nm | 590 nm | 660 nm |
|---|---|---|---|---|---|---|
| BBOA | $Abs_{BBOA}$ (Mm$^{-1}$) | 19.4 | 6.0 | 3.8 | 2.2 | 1.2 |
| | Contribution (%) | 51.9 | 44.5 | 40.8 | 43.4 | 39.1 |
| OOA | $Abs_{OOA}$ (Mm$^{-1}$) | 4.1 | 2.1 | 1.5 | 0.8 | 0.5 |
| | Contribution (%) | 11.1 | 16.5 | 16.9 | 16.6 | 16.6 |
| aqSOA | $Abs_{aqSOA}$ (Mm$^{-1}$) | 6.2 | 2.3 | 1.6 | 0.9 | 0.7 |
| | Contribution (%) | 16.4 | 21.4 | 20.8 | 22.1 | 26.7 |
| CCOA | $Abs_{CCOA}$ (Mm$^{-1}$) | 4.3 | 1.0 | 0.7 | 0.4 | 0.2 |
| | Contribution (%) | 11.5 | 7.8 | 7.5 | 7.4 | 7.5 |
| HOA | $Abs_{HOA}$ (Mm$^{-1}$) | 3.4 | 1.3 | 1.3 | 0.5 | 0.3 |
| | Contribution (%) | 9.1 | 9.8 | 14.0 | 10.5 | 10.1 |

Figure 7: Are the "MAC" values the a-e constants in the MLR method? If yes, then please explain this in the text clearly. Same comment for lines 459-460.

**Response:** Yes. The "MAC" values are the a–e constants in the MLR method. It has been revised in the revised manuscript, which reads "…the a–e were constants, used to optimize the $Abs_\lambda$ of each OA component, and equivalent to MAC values at each wavelength (i.e., a–e at 370 nm represented $MAC_{370,OOA}$, $MAC_{370,BBOA}$, $MAC_{370,CCOA}$, $MAC_{370,aqSOA}$, and $MAC_{370,HOA}$, respectively)." **(Lines 236-239).** We also have revised in **lines 558-559** "The MAC values of the five resolved OA components equivalent to the a–e values in the MLR method at different wavelengths were shown in Fig. S14."

Lines 452-454: Please reformulate this sentence, it is grammatically incorrect.

**Response:** It has been revised accordingly in the revised manuscript, which reads "Compared with other OA factors (except BBOA), the contribution of $Abs_{370,aqSOA}$ to $Abs_{370,BrC}$ was relatively higher (Table S1), when the correlation between $Abs_{370,BrC}$

and aqSOA concentrations was also stronger. These results might be related to the aqSOA formed from the aged BBOA via aqueous-phase reactions." **(Lines 550-553).**

Lines 462-465: Please reformulate this sentence, it is grammatically incorrect.

**Response:** It has been revised accordingly in the revised manuscript, which reads "Previous research found that the MAC of BBOA at 365 nm was twice that of SOA, which was associated with the water-soluble BrC (Lorenzo et al., 2017; Washenfelder et al., 2015)." **(Lines 562-564).**

Lines 465-467: What the values in parenthesis? Are they from the literature? Have you calculated them? If so why mentioning at wavelength 370-660 nm while it is later introduced at 370-880 nm at line 505.

**Response:** The values in parenthesis represented the fitted AAE values of OA factors, that were calculated by a power-law fitting of the absorption coefficient for these OA factors from 370 nm to 660 nm in this study. More details concerning this approach were described in the literature (Qin et al., 2018; Wang et al., 2019). Here, we assumed that $Abs_{880}$ was sole from BC, and Abs can be divided into BC and BrC absorption at 370, 470, 520, 590, and 660 nm. Thus, AAE values of OA factors were calculated from 370 nm to 660 nm (i.e., $AAE_{370-660,aqSOA}$), while AAE values of aerosols were calculated from 370 nm to 880 nm ($AAE_{370-880}$). It has been revised accordingly in the revised manuscript, which reads "The AAE values of OA factors, calculated by a power-law fitting of Abs for these OA factors from 370 nm to 660 nm (Qin et al., 2018; Wang et al., 2019), were shown in Fig. S14. It should be noted that aqSOA had the lowest $AAE_{370-660,aqSOA}$ value (3.54), while BBOA has the highest $AAE_{370-660,BBOA}$ value (4.93)." **(Lines 564-568).**

Lines 467-469: Same comment as for lines 443-445, where do the numbers come from?

**Response:** The values were calculated by a power-law fitting of the absorption coefficient for OA factors from 370 nm to 660 nm in this study. More details concerning this approach were described in the literature (Qin et al., 2018; Wang et al., 2019). We have revised in the revised manuscript: "The AAE values of OA factors, calculated by a power-law fitting of Abs for these OA factors from 370 nm to 660 nm (Qin et al., 2018; Wang et al., 2019), were shown in Fig. S14." **(Lines 564-566).**

Lines 472-482: This entire paragraph is based on discussion of an SI figure (S9). I would suggest moving this figure to the main. Eventually, Fig. 7 could go in the SI as it does not provide much information and is not discussed much.

**Response:** It has been revised accordingly in the revised manuscript.

Line 487: Change "but no OOA" to "this was not the case for OOA".

**Response:** It has been revised accordingly in the revised manuscript.

Lines 488-489: Please reformulate this sentence, it is grammatically incorrect.

**Response:** It has been revised accordingly in the revised manuscript, which reads "The slope of the linear regression (3.50) between aqSOA mass concentrations and $Abs_{370,BrC,sec}$ was higher than OOA (Fig. S16), so was the MAC values of aqSOA across UV to Vis range (Fig. S14)." **(Lines 589-591).**

Line 496: The relation between $Abs_{370,BrC,pri}/\Delta CO$ looks more like a cloud of point to me.

**Response:** It has been revised accordingly in the revised manuscript, which reads "Fig. 8 shows that the values of $Abs_{370,BrC,sec}/\Delta CO$ increased with the increases of aqSOA and ALWC concentrations from 17:00 to 03:00 LT ($r^2 = 0.63$, $0.57$, $p < 0.001$), while $Abs_{370,BrC,pri}/\Delta CO$ slightly decreased with the increases of BBOA and m/z 60 concentrations ($r^2 = 0.35$, $0.33$, $p < 0.001$)." **(Lines 596-599).**

[Figure]

Line 501: Change "should" to "could".

**Response:** It has been revised accordingly in the revised manuscript.

Line 508: I would define a correlation of 0.49 as strong. Even though it remains unclear if $r^2$ is really the correlation or the coefficient of determination.

**Response:** Yes. Here, $r^2$ is a Pearson correlation squared, which has been defined at **line 291-292** "Moreover, BBOA showed significant correlations with m/z 60 (mainly $C_2H_4O_2^+$) and m/z 73 (Pearson's $r^2$ ($r^2$) = 0.85, 0.80, $p < 0.001$)…".

Lines 513-515: I might be mistaken but isn't this relation circular? Since AAE is calculated based on a log relation of Abs values and BC is linearly dependent on Abs, don't we expect the relation we see on Fig. 9?

**Response:** Thanks for the reviewer's comment. While AAE was calculated based on a power-law fitting of absorption values (Qin et al., 2018; Wang et al., 2019), BC concentration is linearly related to $Abs_{BC}$. However, OA concentration does not necessarily show a linear relationship with $Abs_{BrC}$. The mixing state of BC and OA, which depends on combustion conditions, can also influence AAE. Previous studies (Lu et al., 2015; Saleh et al., 2014) have shown that biomass-burning emissions can affect absorption properties, which is reflected in the relationship between AAE and the BC-to-OA ratio. The observed relationship in Fig. 9b is thus not circular but reflects the effect of biomass-burning emissions on aerosol absorption properties. The parameterized curve in our study (black curve) aligns with previous research (red curve) using similar wavelength ranges (370–880 nm) (Lu et al., 2015).

It has been revised accordingly in the revised manuscript, which reads "AAE was calculated using a power-law fitting of aerosol absorption values (Qin et al., 2018; Wang et al., 2019). While BC concentration is linearly dependent on $Abs_{BC}$, OA concentration does not follow the same pattern with $Abs_{BrC}$. The mixing state of BC and OA, influenced by combustion conditions, can also affect AAE. Previous studies have shown that biomass-burning emissions can impact absorption properties, which is reflected in the relationship between AAE and the BC-to-OA ratio (a measure of the combustion conditions) (Lu et al., 2015; Saleh et al., 2014). Thus, the relationship observed in Fig. 9b reflected the influence of biomass-burning emissions during the campaign, and the parameterized curve in this study (black) was consistent with prior

research (red) using wavelengths from 370 nm to 880 nm (Lu et al., 2015)." **(Lines 621-631).**

Lines 526-529: It appreciated to have a concluding statement here, but feels redundant since a conclusion is provided just a few lines after.

**Response:** Thanks for the reviewer's comment. We have revised and moved it to **lines 670-676** "In conclusion, our results revealed the aqSOA formation and brownness from aged BBOA via aqueous-phase reactions and highlighted the importance of aqSOA on aerosol pollution and absorption in the Sichuan Basin, China. Brown aqSOA originating from biomass-burning emissions was an important player in air quality budget and climate forcing balance worldwide. And it should be taken into account in air quality and climate models for a correct description of the global OA budget and its climate-relevant optical properties."

Line 540: I would call this section "Conclusion" and not "Implications" as the content of the section is about summarizing and repeating the content of the manuscript.

**Response:** Thanks for the reviewer's suggestion. It has been revised accordingly in the revised manuscript.

Line 541-542: This sentence is a copy-paste of the first sentence in the intro. Please reformulate or delete.

**Response:** We have deleted this sentence in the revised manuscript.

Line 545: What do you mean by "an increasing laboratory research"?

**Response:** This sentence has been removed in the revised manuscript.

Lines 551-552: You mention that your study highlighted the importance of aqSOA on air quality. However, this is the first time in the paper that air quality is mentioned.

**Response:** Thanks for the reviewer's comment. We have revised and moved it to **lines 670-672** "In conclusion, our results revealed the aqSOA formation and brownness from aged BBOA via aqueous-phase reactions and highlighted the importance of aqSOA on aerosol pollution and absorption in the Sichuan Basin, China."

Lines 552-557: As shown in Fig. 7, isn't it BBOA that has the strongest absorption across UV to visible, rather than aqSOA?

**Response:** Thanks for the reviewer's comment. BBOA has the strongest absorption across UV to visible rather than aqSOA. It has been revised accordingly in the revised manuscript, which reads "The aqSOA formed from aged biomass-burning emissions significantly contributed to the BrC budget and showed stronger absorption across ultraviolet to visible range than other OA components (except BBOA)." **(Lines 662-664).**

Lines 558-559: This sentence is redundant.

**Response:** It has been revised accordingly in the revised manuscript.

Line 560: Saying "considerable" implies that you can quantify it. However, in this manuscript the amount of aqSOA that originates from BBOA is not quantified. Please rephrase.

**Response:** It has been revised accordingly in the revised manuscript.

Line 564: Not sure how Fig. 6 justifies that the formation of aqSOA was through aqueous-phase reactions instead of photo-chemical reactions.

**Response:** Thanks for the reviewer's comment. In the revised manuscript, we have confirmed that aqueous-phase reactions were an important source of aqSOA in **lines 450-461**. Additionally, we further demonstrated that less oxidized SOA during PP was primarily formed through aqueous-phase rather than photochemical reactions in **lines 479-491**.

"As shown in Fig. 6, the bottom region of the triangle was dominated by BBOA, CCOA, and HOA with low $f_{44}$ (0.040, 0.017, and 0.016, respectively) in this study, indicating that they were freshly emitted and less oxidized. However, the $f_{44}$ of SOA factors (i.e., OOA and aqSOA) (0.118 and 0.117) were observably higher than POA factors, showing the freshly oxidized properties of SOA. Meanwhile, $f_{44}$ of aqSOA was close to that observed in fogs (Gilardoni et al., 2016; Kim et al., 2019), highlighting the presence of aqueous-phase reactions in this study. The relative abundance of m/z 45 (mainly $HCO_2^+$), a tracer ion for carboxylic acids, was higher in the aqSOA spectra than in the OOA spectrum (Fig. S9). It was consist with previous research which found that aqueous-phase reactions were important sources of oxygenated organic compounds, including organic acids (Ervens et al., 2011; Kim et al., 2019; McNeill, 2015; Sun et al., 2010; Yu et al., 2014)." **(Lines 450-461).**

"During PP, the $f_{44}$ values ranging from 0.022 to 0.140 (0.080 ± 0.035) were significantly higher than that during CP (0.021−0.150, 0.064 ± 0.019) ($p < 0.001$), while the $f_{43}$ value was slightly lower with an average of 0.062 ± 0.027. Compared with CP ($r^2 = 0.17$, slope = −0.53), $f_{44}$ showed a more significant increase as the decreasing of $f_{43}$ with higher $r^2$ value (0.70) and the regression slope of $f_{44}$ versus $f_{43}$ (−1.09) was closer to −1 during PP. This indicated that more aged SOA existed in the atmosphere during PP (Fig. 6a and c). It should be noted that the points of $f_{44}$ versus

$f_{43}$ were inside the upper boundary of the triangle region, and most points were outside the bottom boundary of the triangle region during PP. These results suggested that less oxidized SOA were formed via aqueous-phase reactions instead of photo-chemical reactions during PP (Kim et al., 2019; Zhao et al., 2019). Moreover, these points outside the bottom boundary of the triangle region with higher $f_{44}$ (> 0.05) and lower $f_{43}$ (< 0.06) showed relatively higher ALWC during PP, but not during CP." **(Lines 479-491).**

Lines 571-574: Please reformulate this sentence, it is grammatically incorrect.

**Response:** Thanks for the reviewer's comment. It has been revised accordingly in the revised manuscript, which reads "Higher values of $AAE_{370-880}$ and $MAC_{\lambda,aqSOA}$ reinforced that aqSOA formation from aged biomass-burning emissions via aqueous-phase reactions had stronger absorption than that via photo-chemically reactions." **(Lines 667-669).**

Lines 575-577: Please reformulate this sentence, it is grammatically incorrect. Seems like part of the sentence is missing.

**Response:** Thanks for the reviewer's comment. It has been revised accordingly in the revised manuscript, which reads "In conclusion, our results revealed the aqSOA formation and brownness from aged BBOA via aqueous-phase reactions and highlighted the importance of aqSOA on aerosol pollution and absorption in the Sichuan Basin, China." **(Lines 670-672).**